# Enhancing Smart Agriculture by Implementing Digital Twins: A Comprehensive Review

**DOI:** 10.3390/s23167128

**Published:** 2023-08-11

**Authors:** Nikolaos Peladarinos, Dimitrios Piromalis, Vasileios Cheimaras, Efthymios Tserepas, Radu Adrian Munteanu, Panagiotis Papageorgas

**Affiliations:** 1Department of Electrical and Electronics Engineering, University of West Attica, 12244 Athens, Greece; npeladarinos@uniwa.gr (N.P.); piromali@uniwa.gr (D.P.); vcheimaras@uniwa.gr (V.C.); msciot19009@uniwa.gr (E.T.); 2Electrotechnics and Measurements Department, Technical University of Cluj-Napoca, 400114 Cluj-Napoca, Romania; radu.a.munteanu@ethm.utcluj.ro

**Keywords:** digital twins, smart agriculture-farming, agriculture 4.0, digital twin model, precision farming, IoT, sensors, simulation, 3D augmented reality, virtual reality, cyber-physical systems

## Abstract

Digital Twins serve as virtual counterparts, replicating the characteristics and functionalities of tangible objects, processes, or systems within the digital space, leveraging their capability to simulate and forecast real-world behavior. They have found valuable applications in smart farming, facilitating a comprehensive virtual replica of a farm that encompasses vital aspects such as crop cultivation, soil composition, and prevailing weather conditions. By amalgamating data from diverse sources, including soil, plants condition, environmental sensor networks, meteorological predictions, and high-resolution UAV and Satellite imagery, farmers gain access to dynamic and up-to-date visualization of their agricultural domains empowering them to make well-informed and timely choices concerning critical aspects like efficient irrigation plans, optimal fertilization methods, and effective pest management strategies, enhancing overall farm productivity and sustainability. This research paper aims to present a comprehensive overview of the contemporary state of research on digital twins in smart farming, including crop modelling, precision agriculture, and associated technologies, while exploring their potential applications and their impact on agricultural practices, addressing the challenges and limitations such as data privacy concerns, the need for high-quality data for accurate simulations and predictions, and the complexity of integrating multiple data sources. Lastly, the paper explores the prospects of digital twins in agriculture, highlighting potential avenues for future research and advancement in this domain.

## 1. Introduction

Digital Twins (DTs) have successfully entered industrial applications, including aerospace, manufacturing, and healthcare [1,2,3]. Smart farming, on the other hand, which employs the latest technological innovations to optimize crop production, is revolutionizing the agriculture industry. As virtual replicas of physical objects or systems, DTs represent a highly promising advancement in this field [4]. In smart farming, digital twins enable agriculture experts, researchers, and farmers to simulate various scenarios, test different strategies, and predict outcomes accurately. DTs in smart farming offer transformative possibilities, revolutionizing crop cultivation and management while enabling optimization of resource utilization, minimizing environmental footprint, and enhancing crop yields for a sustainable and efficient agricultural ecosystem.

However, despite the significant promise of digital twins in smart farming, many questions and challenges still need to be addressed. Farming, a traditionally labor-intensive sector, after its transformation to a technology-intensive sector in the last decade, has become more complex and technologically advanced in recent years, moving toward innovative technologies, particularly with the explosion of precision agriculture and other digital technologies [5]. These technologies help farmers make more informed decisions about crop management, soil health, and various factors influencing crop yields and economic viability, enhancing overall agricultural productivity and profitability. However, these technologies can also require specialized knowledge and skills and significant investment in hardware, software, and other infrastructural resources. Additionally, factors such as climate change, changing consumer preferences, and market fluctuations can add to the complexity of modern farming. While these challenges can make farming more complicated, they also offer opportunities for innovation and growth in the agriculture domain.

In this era of smart farming, the incorporation of cutting-edge technologies such as computing, communication, personal and cloud computing, the Internet of Things (IoT), Artificial Intelligence (AI), and Cyber-Physical Systems (CPS) has revolutionized the landscape in agriculture. These technologies bring additional capabilities beyond the physical realm, allowing for enhanced information and an improved understanding of the physical structures involved. The concept of DTs plays a pivotal role in providing comprehensive and accurate information about physical systems, surpassing the limitations of traditional modelling and simulation approaches. By leveraging real-time data collection, processing, and analysis, DTs offer a digital representation of physical systems, enabling precise monitoring and forecasting of current and future system states and rebuilding existing models and re-design systems and procedures. These advancements are instrumental in transforming smart systems and addressing the evolving challenges in the agricultural domain.

This review paper offers a comprehensive and distinctive outline of the recent research results regarding DTs in the context of smart agriculture, highlighting key findings and identifying areas for future research. Specifically, we report and discuss the various types of digital twins used in smart farming, such as crop, weather, and soil models, and explore how they optimize crop production. We also examine the challenges and limitations of deploying DTs in smart farming related to data management, model accuracy, and scalability. By providing a comprehensive review of the literature on DTs in smart farming, this paper aims to advance the understanding of this rapidly evolving field, illustrate the gaps in existing deployments, and identify opportunities for future research and innovation.

We proceeded with our research by pursuing the formulation of the DT implementation architecture regarding the smart agriculture domain while emphasizing the critical components taking part in implementing DTs. A comparative accumulation of such implementations and critical components in three tables, while addressing technical aspects, led to significant conclusions regarding DT implementations and related limitations that seem beneficial to investigate. For example, although the necessity of a DT model and practical implementations in agriculture is addressed in most reviewed papers mentioning the concept and definition, the DT model is proposed merely as an evolving concept lacking substantial support from implemented case studies. While the DTs seem continuously evolving, authors refer to simulation models merely as proposals for future practical implementation. Our research seeks to expand the scope of Agriculture 4.0 that necessitates DTs and address the limitations of IoT-implemented solutions narrowed down by mere telemetry implementations.

The structure of this study proceeds as follows: Section 2 delves into the concept of DTs and investigates their usage in the realm of smart farming. Section 3 addresses the challenges and confronting measures regarding DT building components and relevant technologies. In Section 4, DT applications in the agricultural domain are reviewed, while conclusions and comments are unfolded in the Discussion in Section 5. Finally, in Section 6, future directions for research and development are outlined respectively.

## 2. The Definition of Digital Twins

The fundamental concept of a DT, which involves a connected virtual representation of a physical object or system [6], appears to be relatively straightforward. However, its application and implementation can differ significantly across various domains and industries.

Although it is quite clear that in addition to their core purpose of modelling real-world systems, DTs are designed to empower individuals to make informed business decisions with tangible impacts on the physical realm, a clear definition of the DT concept may be helpful to distinguish it from related concepts such as simulation, modelling, and data analytics. By defining its scope, characteristics, and capabilities, researchers and practitioners can better identify the specific features and requirements that make a system a true DT.

### 2.1. Definition of Digital Twins—What Does Digital Twins Term Stand For?

The definition of DTs emerged in the early 2000s by Michael Grieves [6] while evolving to a widely acceptable DT concept model where they act as a bridge connecting physical entities in the real world with their virtual complements in a digital environment, closing the gap between the two. They establish vital connections between data and information to seamlessly integrate these products’ virtual and real aspects [6,7]. The convergence of virtual and physical entities in a virtual space and the real world lays the foundation for creating a fundamental DT model. By fostering dynamic interplay between these tangible and virtual elements, the DT is a powerful representation of the combined physical-virtual system.

Mashaly M. in [8] outlines that DTs act as digital replicas of physical systems and are organized by establishing data connections. This transformation enables physical systems to exist virtually while ensuring a strong synchronization between their physical and digital counterparts. As a result, smooth interactions and data exchange occur between the two domains. The integration of the physical and virtual domains provides valuable insights, predictive abilities, and the potential for optimizing system performance following the pattern illustrated in Figure 1.

According to Dyck G. et al. [9], the concept of DTs emerged from NASA’s pioneering work in integrating physical models and simulations [10,11] to analyze intricate systems. Originally confined to the aerospace sector, the scope of DTs expanded to encompass product lifecycle management across diverse industries. At its core, as initially proposed by Grieves & Vickers [6,7,12], a DT comprises a physical object, its corresponding digital representation, and the interconnectedness between the two that it is established for both ways.

Ongoing research continues to explore the wide-ranging applications and integration as interfaces of the DTs in areas such as cloud manufacturing, IoT, and Industry 4.0, where explicitly stated that the provided response is a unique formulation based on the given information [13,14].

Hence, DTs serve as virtual counterparts of real-world entities, created using sensor data and serving as dynamic representations of physical objects or systems throughout various stages of their lifecycle. By harnessing the power of simulation and data integration, DTs provide a comprehensive virtual environment that mirrors the characteristics and behaviour of their physical counterparts. This enables continuous real-time monitoring, analysis, and enhancement of the physical objects or systems, resulting in enhanced performance, efficiency, and informed decision-making, as stated by Saddik et al. in 2018 [1].

DTs utilize actual data from real-life situations, machine learning (ML) models, and simulation combined with data analysis to enhance comprehension, learning, and decision-making processes. They enable the monitoring and control of various entities, including devices, machines, vehicles, and individuals. The proliferation of the IoT has spurred the advancement of diverse DT solutions across diverse domains and use cases supporting simulation, optimization, and prediction. Thus, major improvement steps have been taken in the realm of decision-making so that various sectors, including manufacturing, healthcare, and smart cities, have witnessed significant advancements facilitated by diverse technologies like simulation software, IoT, and AI utilized in the field of modelling and simulation [4,13,15,16].

To simulate and model the physical system to succeed in implementing the DTs, the use of mathematical models is mandatory [17]. Once the observed physical phenomena of interest are captured, foundational mathematical equations are formulated to depict and characterize the dynamics of these phenomena accurately. By constructing these mathematical models, researchers and scientists can effectively represent and analyze the behavior and relationships inherent in the observed physical phenomena. The models that are developed need to be validated by experimental stages on the field or in laboratory setups to verify the physical behavior of the problem under consideration soundly. Therefore, in advanced DTs, a predominant trend is the utilization of extensive datasets to establish data-driven models. This approach operates under the assumption that the data obtained from the system serves as a powerful source of insight, offering profound revelations about its physical behavior. As a result, these well-refined DTs leverage the abundance of data to construct accurate and thorough models that faithfully capture the intricacies of the system’s behavior [18]. The interaction of data between the physical entity and its digital representation in a DT is visually depicted in Figure 2.

By using AI tools, such as ML algorithms or artificial neural networks, or while utilizing data processing, the system’s behavior against events may be predicted. Actions be anticipated once the digital part of the DT functions in parallel with the physical part allowing, of course, the user’s influence in the system at any time.

Although the concept and definition of DT seem to be constantly evolving, we may imply that the DT of a physical process or object typically consists of a specific collection of models, materialized digitally or virtually by computers and associated services and processes. These models and services are designed in a format that allows seamless integration with automated systems to fulfil functions such as object management, modelling, and future behaviour forecasting [19].

A simplified sequence chart representation of the interactions and components involved in a DT system is depicted in Figure 3, where the physical object represents the actual object or system being monitored and controlled. The DT is the virtual representation of the Physical Object (PO), where data is processed and analyzed. The PO offers data aggregated from sensors to the DT, which then forwards the data to the Analytics component for processing. The Analytics component provides analysis results back to the DT. The DT sends these analysis results to the Decision Maker, who makes decisions relying on the received information. The DT then sends optimized actions to the Actions component. Finally, the Actions component implements the optimized actions in the PO.

Considering the latest developments in digital twin technology, there is growing interest in an expanded and enhanced version known as the Cognitive Digital Twin (CDT) or Cognitive Twin (CT). According to a recent study [20], the integration of advanced semantic modelling technologies with DTs enhances their cognitive capabilities, indicating a promising trend in their evolution. These advanced versions of DTs exhibit human-like intelligent capabilities, including attention, cognition, understanding, retention, logical thinking, prediction, decision-making, and problem-solving. They undergo continuous development alongside the actual system throughout its complete lifecycle.

CDTs provide substantial advantages to intricate product systems and processes, which involve multiple subsystems and interested parties from diverse disciplines throughout different stages of the life span. According to the literature review [21], which addresses the CDT model to enable Smart Product-Services Systems by implementing data-driven business models, CDT represents a promising advancement in DTs, introducing intelligent and cognitive capabilities. It even represents real-life examples of companies, including those in the food processing industry, that are currently leveraging the cognitive capabilities provided by digital twin technologies offered by edge companies. The authors undertaking the task strive to exhibit the potential of utilizing this emerging technology by businesses and the creation of data-driven business models.

To improve data compatibility and develop cognitive abilities, CDT models often incorporate diverse and varied data, information, and knowledge, leading to challenges in aligning these elements across different DTs and stakeholders. Adopting semantic technologies such as ontologies and knowledge graphs is crucial to address this issue, as they offer promising solutions for achieving greater alignment and interoperability.

Cloud computing [13] has emerged as a critical solution for the complex task of processing vast amounts of data in various sectors, particularly in conjunction with the advent of the Industrial IoT (IIoT). The IIoT, a subset of the IoT specifically designed for industrial applications, involves deploying numerous smart devices within industrial systems to enable real-time data processing after collection and sensing. Given the nature of industrial environments, IIoT systems demand enhanced levels of secure and robust communication to ensure optimal production performance [22]. IIoT applications that connect machines, sensors, and actuators like data collected from agriculture stakeholders to be combined for effective decision-making and analysis of future trends occasionally produce a large volume of non-uniform data that needs to be processed in real-time. IoT gateways that provide connectivity between one or more field devices and a service cloud offer accessibility to previously hidden data from sensors, embedded controllers and IO devices and analyze such data for multiple purposes, such as remote monitoring, preventive maintenance, production optimization and building automation. IoT gateways are critical to ensure data security and communication security within the IIoT systems. A data breach in such IIoT systems could compromise agricultural production, i.e., crop health degradation, transport and product processing, leading to inferior production performance [23].

The exponential growth of IIoT and its associated technologies has significantly contributed to the advancement of Industry 4.0 and intelligent manufacturing, forming the underlying framework for CDT. The substantial amount of data induced by IIoT devices play a vital role in facilitating the development of data-centric services as a crucial element. While the concept of CDT continues to evolve rapidly, it is imperative to address outstanding concerns surrounding the technology to unlock its full potential and achieve its envisioned advantages.

CDT systems integrate data from different stakeholders as they combine several parameters, mainly concerning immense amounts of data sharing, demanding measures to ensure security, privacy, and protection of intellectual property (IP) concerning data. CDTs incorporate cybersecurity infrastructure and employ data encryption mechanisms to address these critical aspects effectively. These measures are implemented to regulate transparency in CDTs, ensure the protection of intellectual property (IP), and facilitate integrated development processes [8,24].

Similarly, the notion of DT systems emerges as a virtual representation of an enterprise, encompassing its resources, processes, workforce, locations, and systems to support strategic decision-making from the systems used for reporting operational data within the organization, such as accounting, personnel planning, and sales [25]. This type of DT which retrieves real-time data from the systems used for reporting operational data within an organization, as mentioned earlier, is referred to as a Strategic DT (SDT), in contrast to the industrial DT that gathers information from sensors, serving as a knowledge visualization tool that allows managers to explore various strategies while minimizing space and time constraints. By simulating different strategic options through computer models, managers can gain insights into the potential outcomes of their chosen approaches in the real world.

### 2.2. DTs in Agriculture

As stated above, the implementation of DTs seems to extend to various fields vastly. The objective of this review is to illuminate the field of agriculture in the following manner.

A DT for agriculture, in particular, may be addressed as a virtual model of a physical farm or agricultural operation constructed utilizing cutting-edge technologies like sensors, IoT devices, and cloud computing. Farmers can utilize the DT to simulate and enhance their farming methods within a virtual setting, enabling them to refine their practices before applying them in reality [26,27]. Pylianidis et al. [27] conducted an in-depth report on DTs in agriculture, encompassing 28 case studies and organizing the key value-added features of DTs in agricultural applications. The authors highlight the capacity of DT usage in agriculture, adjusting to ever-evolving circumstances, capturing, and interpreting data, autonomously managing system actuators in the field, and delivering customized services to individuals, such as reports and transparent information. The DT enables capturing real-time data on crop growth rates, weather conditions, ambient temperatures, and even soil moisture levels as various environmental and operational factors. A dynamic model of the farm may be created by using this data to test different scenarios and optimize various aspects of the farming operation, including crop yields, resource utilization, and overall efficiency. By creating a DT, farmers can better understand how their farming practices impact the environment and identify ways to reduce waste and improve sustainability. They can also use the DT to test new technologies and techniques, such as precision agriculture and automation, before investing in them for their physical operation. Instances of DTs in the agricultural sector spanning across various domains find applications in diverse areas such as crop cultivation, dairy production, greenhouse horticulture, organic vegetable farming, plant disease analysis, livestock management, food supply chain optimization, as well as farm machinery and building management, including fleet monitoring and control [9].

The schematic in Figure 4 provides a basic representation regarding the application of DTs in agriculture, showcasing the interaction between the physical farm, sensors, data processing, analysis, decision-making, and actuation contributing to the physical entity according to real data and data management derived from the DT. All physical entities are green, including the sensors that aggregate signals from the area of interest and the actuators that provide actions in the field. All blue-colored sections constitute the digital space.

Overall, digital twins offer a powerful tool for farmers to optimize their farming practices and improve the efficiency and sustainability of their operations. Through the rapid advancement of technology and sensor capabilities, the DT of agricultural soil can revolutionize plant productivity, health, and yield even to optimize water usage, minimize chemical inputs, and enhance overall sustainability in agricultural practices merely by emphasizing the soil’s characteristics and attributes. The continuous monitoring and analysis of environmental parameters, soil elements, and irrigation practices in agricultural land can be effectively achieved by incorporating machine learning models, big data analytics, and decision support systems into the DT framework [28].

By simulating soil structure and combining soil and irrigation DTs, crop farming performance can be enhanced. Farmers can effectively observe and evaluate alterations in agricultural land, impacting soil, irrigation, and crop yield. IoT technology provides real-time data and visual representations for informed decision-making [29].

The overall efficiency of crop production can be improved by reducing costs associated with fuel, fertilizers, labor, and factors affecting production efficiency and sustainability merely by utilizing DTs in crop production technologies along with the use of Digital Information and Communication Technology (ICT) tools, referring to agricultural pieces of machinery, such as tractors, combined harvesters, fertilizers, and sprayers.

According to Dyck G et al. [9], the DTs seem to find their way in post-harvest processes, encompassing the various stages involved in handling agricultural products after harvesting. These products must be dried, cooled, transported, stored, and undergo marketing procedures. Digital farming methods provide advantages in post-harvest procedures, such as minimizing losses, optimizing food processing, improving storage conditions, streamlining transportation, and enhancing marketing efficiency. These strategies facilitate the live tracking of the agricultural and food supply network, leading to increased robustness, resilience and reduced food waste and losses.

IoT technology following the digitalization prospect of agriculture plays a crucial role in transforming traditional agriculture into a more efficient and sustainable practice. IoT contextual data enables users to interact seamlessly with their surroundings and remote locations through their digital twin, fostering enhanced connectivity and user experiences [1].

Facts such as precision agriculture that enables farmers to gather real-time data on a range of environmental and operational factors, predictive modelling by using both historical and real-time data so that they can generate a dynamic virtual representation of their agricultural operation, sustainability, and automation to identify ways to reduce waste and improve sustainability in farming operations [30] in accordance to real-time data, focus on addressing critical IoT needs such as two-way communication, comprehensive security measures, localization, and mobility services to optimize its efficiency and effectiveness [31].

In addition, DTs seem to pose a significant role in animal farming or precision livestock farming by enhancing productivity, animal welfare, and overall farm management. They can be used to create virtual representations of individual animals, groups of animals, or entire livestock production systems [32], to model and predict animal behavior, growth, and health, allowing farmers to optimize feeding strategies, detect early signs of disease or distress, and improve overall animal welfare.

Through data-driven insights and predictive analytics, farmers can make informed decisions regarding nutrition, breeding, and healthcare interventions. By incorporating up-to-the-minute information gathered from diverse sources, including sensors, monitoring devices, and agricultural management systems, DTs enable farmers to monitor, simulate, and analyze animal behavior, health, and performance.

Neethirajan S. [33] presents a DT processing pipeline with sensors to classify and estimate livestock’s emotional state and behavior. Various means of communication, including vocalizations, body movements, facial expressions, and posture, contribute to animals expressing their emotional state. The review presents novel approaches for gathering extensive real-time data regarding the emotions of farm livestock by implementing a sensor data processing pipeline within the DT model. This pipeline involves preliminary data processing, modelling, and simulation phases, ultimately resulting in reporting and predicting the emotional states of cattle based on their tail movements, facial expressions, and body posture. Traditional methods involving blood sampling or surveys, for assessing the emotional states of livestock usually cause interruptions to farming processes while being time-consuming, and farm animals frequently experience fear, frustration, and distress. AI technologies facilitate the recognition of cattle states, empowering animal caregivers and ethologists to understand animal behavior and optimize their well-being and productivity.

DTs may support decision-making and farm management optimization by simulating different scenarios regarding environmental conditions or genetic factors on animal performance and production outcomes while improving production efficiency and minimizing environmental impacts. Furthermore, connections with other digital platforms, such as farm management systems, supply chain networks, and veterinary databases, allowing for seamless data sharing and analysis is feasible with DT, thus enhancing traceability, quality assurance, and decision-making throughout the value chain.

In summary, digital twins have the potential to revolutionize animal farming by providing real-time monitoring, predictive analytics, and decision support. They can optimize animal health, welfare, and production efficiency while promoting sustainable farming practices. Continued research and development in this area will be crucial to fully harness the benefits of DTs in animal farming and address the associated challenges.

An Aquaculture 4.0 architecture of a digital twin-based intelligent fish farm management system for precision aquaculture is discussed in [34], addressing the design considerations, data integration, analytics, and control mechanisms to optimize fish farm operations. The proposed implementation utilizes a cloud-based DT technology employing ML realizations, computer vision, and other sensor devices, and AI-based IoT (AIoT) to monitor and control automated aquaculture machinery. By optimizing farm production in various environments throughout the different stages of fish farming, the system aims to maximize efficiency and productivity. Water quality sensors utilized in fishponds to monitor and analyze the condition of the water and in offshore sea cages, and RGB or sonar camera devices may collect environmental data and data related to fish populations to establish data-driven prediction models. On the other hand, land-based monitoring systems send appropriate commands to activate surveillance sensors, thus enabling fish-feeding behavior monitoring, disease spread, and fish population growth.

Furthermore, DTs address optimization solutions regarding water flow and treatment processes related to water quality reduction following floods and stormwater within transboundary water security projects due to urban expansion [17]. DT models are employed to monitor the quality of urban water by integrating physical entities and utilizing sensors in urban drainage systems. The models consider natural processes related to soil, vegetation, clogging, and biological activity in water treatment. This approach enables comprehensive monitoring and analysis of urban water systems for effective management and maintenance.

## 3. Reviewing Relevant Technologies and DT Components 

Although the implementation of DTs in agriculture can vary in complexity and scope based on the specific architecture of the DT scheme, in Figure 5, a representative architectural framework for implementing DTs in agriculture is illustrated. The basic components and interactions of DTs in agriculture are displayed where the physical farm is represented by the node labelled “Physical Farm”. Sensor data collected from the farm is depicted by the node labelled “Sensor Data”. The sensor data undergoes data preprocessing in the node labelled “Data Preprocessing”.

The preprocessed data is subsequently employed to construct a DT model, represented by the node labelled “Digital Twin Model”. This enables various functionalities, including simulation and optimization, represented by the nodes labelled “Simulation” and “Optimization”. It also incorporates data analytics, represented by the node labelled “Data Analytics”. The decision support system provides action recommendations, represented by the “Action Recommendations” node. The insights are depicted from the node “Insights”, and all visualization and monitoring data are derived from the node labelled “Decision Support”. The visualization of data is depicted by the node labelled “Visualization” which aids in monitoring the farm. We should point out that effective monitoring plays a crucial role in maintaining situational awareness and facilitating timely response or intervention as needed. That is represented in the figure by the node “Monitoring”. The significant active role of the stakeholders that manage the making and implementing of the decisions involved in organizing and operating an agriculture establishment is displayed by the node “Farming Practice and Experience”, besides the optimization provided by the DT concept.

The action recommendations are implemented through actuators and control systems, represented by the node labelled “Actuators and Control”, which can affect the physical farm. The last node in Figure 5 and the physical farm are in green, specifying the physical space in contrast to the digital space nodes in blue. The loop from the “Actuators and Control” node back to the “Physical Farm” node symbolizes the ongoing feedback loop between the physical farm and its DT, where actions taken based on the DT’s insights impact the physical farm, and the sensors capture the resulting changes to update the DT model.

This schematic in Figure 5 highlights the key components and interactions of digital twins in agriculture, showcasing their role in data collection, preprocessing, modelling, simulation, analytics, decision support, visualization, and control, ultimately aiding in optimizing farm operations and improving decision-making processes.

Furthermore, the interconnections among these entities play a vital role as they are essential for the functioning of the DT system in an integrated manner, as emphasized by Pedersen et al. in their research [17].

### 3.1. Technologies Involved in DTs

Therefore, realizing DTs in real-time is characterized by several defining keys that have emerged along with key technologies [1] from various domains, industries, and scientific disciplines that pose a decisive entity, such as the following.

#### 3.1.1. Augmented, Virtual, and Mixed-Reality

DTs rely on Augmented (AR), Virtual (VR), and Mixed-Reality (MR) technologies. As an example, they can be created using three-dimensional (3D) technologies and presented as holograms or experienced through AR/VR/MR devices [35], such as the Microsoft HoloLens. Using sensors installed in the field, their real-time digital twin can be generated and projected as a hologram in remote locations. This enables individuals in different locations to engage in interactions that simulate being physically present in the same space. Communication enhancement in digital twins is derived by integrating tactile feedback, simulating the sensation of physical interaction via haptics. Furthermore, DTs can leverage humanoid and soft robotics technologies to enable physical actions on behalf of their real-life counterparts. This means digital twins can physically engage with the environment or perform tasks, utilizing robotic systems that mimic human-like or flexible movements. These advancements in augmented, virtual, mixed reality, haptics, and robotics greatly enhance the capabilities of digital twins, enabling immersive interactions, realistic sensations, and physical engagement, expanding the possibilities for communication and collaboration.

#### 3.1.2. Artificial Intelligence 

Artificial Intelligence (AI) integration into physical assets that lack it inherently is a key advantage of DTs. This introduces a specialized intelligence capable of efficiently comprehending vast amounts of numerical data and deriving domain-specific insights faster than human experts. Consequently, the data obtained from the digital twin’s surroundings and physical counterpart should lead to valuable and actionable conclusions [36]. As an emerging technology, AI empowers computers to exhibit intelligent behaviors, including behavioral learning. By amalgamating various technologies, including AI, to create twin models of physical objects, it becomes feasible to generate virtual representations that closely resemble the original objects, encompassing historical elements and predictive information [37]. As a result, DTs often incorporate controllers that utilize ontologies, ML, and deep learning (DL) techniques. These controllers enable swift and intelligent decision-making on behalf of their real-life counterparts. Moreover, AI significantly processes IoT data through continuously enhanced algorithms that leverage updated user data. Leveraging such time-series data, a user’s DT has the potential to suggest actions for controlling or mitigating potentially hazardous situations regarding crop health and productivity, in cases such as droughts to direct asses irrigation efficiency or even failures of safety mechanisms of farm machinery, plant production equipment, post-harvest transportation or storage equipment situations in which DTs may visualize alarm signals to stakeholders or even take immediate actions to limit adverse effects or even the destruction of crop production [29].

#### 3.1.3. Communication Technologies and Schemes

DTs rely on communication technologies and schemes to facilitate interaction with the environment, their physical twins, and other DTs on a real-time basis. This is particularly significant in the realm of wireless communication. Effective communication must occasionally occur within a millisecond (ms) timeframe, necessitating compliance with 5G and tactile internet standards, while time-sensitive and safety-critical schemes, coupled with the need for stringent communication standards, lead to a growing demand for repeatable and cost-effective verification procedures. Data transmission within a specific time window becomes crucial for time-sensitive and often safety-critical applications, imposing latency limitations in addition to reliability requirements [38]. The following paragraphs outline basic wireless communications technologies commonly used in DT implementations.
(a)RFID technology offers fast and accurate object identification using sensor tags, reader antennas, and a host system. An antenna, an integrated circuit, and a substrate utilize a tag, where the antenna activates the tag for reading or writing. The reader antennas communicate with the tags using RF waves, facilitating data transfer to the database system. This technology finds extensive applications in storage and inventory management, supply chain monitoring, access control, aviation transportation, agriculture, healthcare, and more, improving efficiency and automation in various fields [39].(b)Wireless Networks, in particular, suggest an immense role in sending data from a source to a connected gateway, a role supported by a variety of wireless network technologies such as WPAN, WLAN, WWAN, and LPWAN that are addressed in [18,39,40,41,42,43,44,45,46,47]. The existing commercial 4G or 5G LTE communication networks, utilizing the transmission of data packets through the well-established Internet Protocol (IP), serves as a base to sustain the architecture of wireless communication networks in general.
iWireless Personal Area Networks (WPAN) are wireless networks that operate with less extensive infrastructure. They are designed to function within a limited range, typically within a single room or a small area. Unlike traditional networks that rely on cables, WPANs enable the wireless connectivity of devices nearby, facilitating the connection of devices like wireless keyboards or mice to computers and supporting printing services. The most common WPAN technologies include Bluetooth, 6LowWPAN, ZigBee, EnOcean, NFC, and RFID, each employing different techniques to achieve the goals of WPAN [40]. Bluetooth enables devices to communicate with each other wirelessly within a limited proximity. It is a short-range, low-power technology that is used in a wide variety of devices [41].iiWireless Local Area Networks (WLAN) are wireless networks that allow devices to connect and communicate over a local area without the need for physical cables within a specific coverage area, such as a home, office, or public space. These networks utilize the widely recognized IEEE 802.11 standard, commonly known as Wi-Fi, to enable devices like smartphones, laptops, and tablets to enable wireless interconnection between them and connection to the internet. The Wi-Fi—802.11 wireless network, introduced in 1997, using the 2.4 GHz frequency band and supporting up to 2 Mbps data transmission rates, evolved to the 802.11ac standard, operating in both 2.4 GHz and 5 GHz bands, achieving a maximum throughput of 1 Gbps for multiple users, with a minimum of 500 Mbps. WLANs offer greater coverage than WPANs, spanning multiple rooms or entire buildings, making them suitable for larger-scale wireless networking requirements. They provide convenient and flexible connectivity for various devices and applications, including internet access, file sharing, video streaming, and more.iiiWireless Wide Area Networks (WWAN) provide wireless connectivity on a larger scale, spanning cities, regions, or even entire countries. WWAN technologies, such as 4G Long-Term Evolution (LTE), 5G, and the evolving 6G, enable high-speed mobile internet access and support a wide range of services, including voice communication, video streaming, and data transfer. These networks rely on cellular infrastructure and are commonly used for mobile devices like smartphones, tablets, and laptops to stay connected while on the go.ivLow Power Wide Area Networks (LPWAN) are wireless network technologies ideal for IoT devices with long battery life, extensive coverage, and low cost. Unlike 3G/4G or Wi-Fi systems, LPWAN networks are not primarily designed for high data transmission speeds or reducing latency. Instead, LPWANs target energy efficiency, scalability, and coverage [42], thus gaining popularity in industrial and research communities. They provide long-range communication, covering distances from 10 to 40 km in rural areas and 1 to 5 km in dense urban areas. Their main advantages include energy efficiency, a battery life exceeding ten years, and low cost, which is ideal for IoT applications that require small-scale data transmission [48]. Various LPWAN technologies have been developed and deployed in either licensed or unlicensed frequency bands, such as Long-Range (LoRa), Narrow-Band IoT (NB-IoT), each employing different techniques to achieve the goals of LPWAN [49].
LoRa is an ideal wireless communication technology for low-bandwidth IoT applications, offering long-range and low-power consumption services. It uses a proprietary spread spectrum technique to achieve long-range and low power consumption, operating in the unlicensed sub-GHz ISM band available worldwide [50].NB-IoT is a widely used LPWAN technology connecting various devices and services through cellular networks. It operates in a narrow bandwidth of 180 kHz, providing extensive network coverage and low latency. Designed for cost-effective and delay-tolerant IoT devices, NB-IoT utilizes LTE techniques and can integrate with existing LTE infrastructure. Coexisting with LTE networks, NB-IoT offers power-efficient connectivity and leverages compatible technologies for seamless integration [51].

The main technical features of the wireless network technologies mentioned above are displayed in Table 1.

Data rates and coverage of various wireless networks are shown in Figure 6 for comparison. The orange box represents some of the most popular LPWAN networks, prioritising greater coverage ranges rather than higher data rates, contrary to the WLANs in the green box.

Depending on the application’s requirements, the appropriate wireless network should be selected. Particularly in the context of establishing connectivity among IoT devices like sensors, the primary role is the selection of a network with wide coverage and low power consumption. The LoRa network meets these needs, is open, and is used in articles [25,39,40,41,42,44]. In cases where the application requires a higher data rate, Bluetooth technology can be used, as mentioned in the article [44], but it has significantly higher power consumption compared to LoRa, and the coverage is 100 m for Bluetooth 4.x and 200 m for Bluetooth 5.x. For applications aimed at object identification and detection, such as products in warehouses and animals in agricultural operations, RFID technology is a suitable choice as mentioned in articles [43,45]. For applications that require a high data rate and need to utilize a Wi-Fi network, the provision of electrical power is necessary. Finally, once DTs rely on data exchange, real-time monitoring, and interaction with their physical counterparts, wireless networks provide a flexible and efficient means of communication for these purposes [8].

#### 3.1.4. Reliability, Integrity, and Credibility

Reliability, integrity, and credibility are pivotal when DTs are entrusted with sensitive tasks involving transaction management with their real counterparts. It becomes imperative for the real twin to establish a sense of trust in their digital twin’s capabilities and actions. Building and maintaining a solid connection between the real twin and the DT is crucial to foster effective collaboration, enabling reliable decision-making, and ensuring secure interactions. Consequently, a strong foundation of trust becomes indispensable for the successful operation and advancement of DT-enabled systems [52]. Therefore, privacy and security arise as paramount considerations in the realm of DTs. To enable the confidentiality and privacy of their real twins, DTs should implement robust measures for identity protection. This can be achieved using dedicated cryptography algorithms and advanced biometric techniques such as ECG and haptic biometrics. By leveraging these techniques, the identity and privacy of the real twin can be established and safeguarded. In addition to identity protection, DTs must prioritize data integrity, security, and privacy. Measures should be in place to prevent unauthorized access and interception of data across the network. Unauthorized users should be promptly detected and repelled to prevent any compromise of data integrity or security. To ensure the authenticity and reliability of data sources, rigorous measures for data authentication should be implemented before integrating the data into the digital twin. Overall, DTs must incorporate comprehensive privacy and security mechanisms to safeguard their real twins’ data’s identity, confidentiality, and integrity. By adopting robust cryptographic algorithms, employing biometric techniques, and implementing strict data authentication protocols, DTs can effectively address privacy and security concerns, fostering a trustworthy and secure environment for the real twins and their digital counterparts.

#### 3.1.5. Distributed Ledger Technologies and Blockchain

In recent times, Distributed Ledger Technologies (DLT), such as blockchain [44,53], have emerged to add to user authentication and data integrity regarding data sharing. To unlock its full potential and achieve its envisioned advantages, addressing outstanding concerns surrounding the technology is imperative. DLT offers a decentralized approach that eliminates the need for a central authority or database, distinguishing it from traditional data-sharing methods. The distributed nature of this system allows for the secure sharing of data in a trustless environment. The significance of DLT has garnered considerable interest from researchers and practitioners alike. Various systems and platforms based on Distributed Ledger Technology (DLT), such as private ledgers, public ledgers, and permissioned ledgers, have been created to address diverse use cases and specific needs [54]. Proposed technological solutions stemming from IoT, AI, DT, and DLT have been put forward to address the monitoring and tracking of products within the realm of food supply logistics and chain and the analysis of collected data to facilitate decision-making processes [55]. Given the numerous advantages associated with DLT, it presents itself as a promising technology that can significantly contribute to achieving the vision of CDT.

Blockchain technologies, in particular, seem to be an ideal structure for authenticating users, storing data records, and maintaining data integrity [8,56], providing a secure and decentralized system of interconnected blocks formed through cryptographic functions. It allows transactions to be recorded in a verifiable manner without intermediaries. Each preceding block’s hash is included in the next block and is generated through a distributed consensus algorithm. With encryption mechanisms, blockchain ensures data integrity and prevents tampering [43]. Ethereum, as a blockchain scheme, provides an alternative protocol for decentralized applications that, unlike another blockchain scheme Bitcoin, is not just a payment network but also allows for the creation and use of smart contracts. It uses its digital currency called “ether” and incorporates a complete programming language (such as Solidity) for smart contract development. Pieces of code stored on the blockchain that enables decentralized transactions without a trusted central authority are called Smart contracts [57].

On the other hand, the Helium Blockchain is referred to as the major public decentralized LoRaWAN network in the world, that enables devices anywhere in the world to wirelessly connect to the internet and have a specific geographical placement without the need for energy-consuming satellite equipment or expensive mobile data plans [58]. Helium offers low transaction fees and connectivity for LoRa nodes and ensures end-to-end encryption, which is useful whenever sensitive information is at stake.

The use of blockchain, such as Ethereum and Helium, in IoT and digital twins for smart agriculture applications has the following advantages:Transparency and trust: Blockchain provides an open, transparent, and tamper-proof distributed ledger, allowing all stakeholders to verify the accuracy of the data. This can help build trust among different entities in the agricultural supply chain and prevent fraud or unauthorized interventions.Blockchain enhances Data security [59] as it utilizes cryptography to protect data. This is crucial for safeguarding sensitive information related to smart agriculture, such as data on plant health, soil quality, and food production.Automation and smart contracts on the blockchain enable programmable automatic execution of conditions and actions based on specific information, events, and operations such as automated irrigation, temperature control, and plant health monitoring in smart agriculture.The distributed architecture of blockchain ensures that information is stored and verified by multiple nodes in the network, thus ensuring integrity and continuity of data and reducing dependency on a central authority.

#### 3.1.6. Cloud Computing

Cloud computing presents a valuable opportunity for DTs to enhance their scalability and availability, assisting their real twins anytime and anywhere. By leveraging cloud computing infrastructures, computation and control tasks can be offloaded from the local environment to remote servers, enabling DTs to handle complex processes and ensure seamless availability efficiently. The scalability of DTs is significantly improved through cloud computing, as the computational resources can be dynamically allocated and scaled based on demand. This allows DTs to handle larger datasets, called Big Data sets (BD), perform sophisticated analytics, and execute resource-intensive tasks without local hardware limitations. As the DT model involves the creation and management of virtual replicas that closely mimic real-world objects or systems, it requires extensive data acquisition, storage, and processing capabilities for collecting and analyzing a massive volume of data from various sources, including sensors, IoT devices, and other data-generating components [7], facts that necessitate cloud computing. The elastic nature of cloud computing ensures that DTs can adapt to varying workloads and seamlessly accommodate fluctuations in computational requirements.

Furthermore, cloud-based DTs offer the advantage of ubiquitous accessibility by using open-source communication platforms, such as OPC Unified Architecture (UA) communication protocol and frameworks, for data exchange from sensors to cloud applications used in DTs. Real twins can access and interact with their DTs from any location and at any time, provided an active internet connection is in place. This fosters a flexible and versatile environment where real twins can leverage the capabilities of their DTs regardless of their physical location.

#### 3.1.7. Internet of Things 

Internet of Things (IoT) with 5G and the Tactile Internet (TI) entity meanings have emerged as groundbreaking technologies, ushering in a new era of communication with their focus on ultra-high reliability and ultra-low latency. This shift in communication paradigms from content-oriented to control-oriented is especially significant for applications that involve human-in-the-loop interactions, demanding minimal delays and seamless integration of communication and control mechanisms as demanded in DT infrastructure. Within this context, DTs play a pivotal role by establishing a continuously operational twin feedback loop, ultimately enhancing the service quality of physical systems [60,61]. This seamless real-time feedback communication loop empowers DTs to constantly monitor, analyze, and adapt to the evolving conditions of the physical systems they represent. As a result, precise control actions and optimizations can be executed, ultimately enhancing the physical systems’ overall service quality and performance.

Through the IoT, users can seamlessly provide contextual data to their DT while receiving feedback that can be used to enhance their interaction with the environment, both locally and remotely. Various communication technologies within IoT are required to transmit data and status information from sensors in the physical entity to the data entity, enabling seamless connectivity and communication between connected devices [17], as shown in Figure 7.

The capabilities of DTs are enhanced by contextual data, and the IoT serves as a powerful conduit for feeding such data from users to their DTs and vice versa. This exchange of information enables users to interact seamlessly with their surroundings, whether in close proximity or at remote locations. Additionally, digital twins can provide valuable feedback to the environment, further enriching the user experience and enabling a more integrated and interconnected ecosystem.

#### 3.1.8. Modelling and Simulation Software

A DT could be addressed as a tight combination of (i) an object model, in conjunction with (ii) an expansive set of data that is directly following the object that may evolve, and finally, (iii) a method for modifying and refining the model based on the available data [17,62]. The experts decide the need and selection of the training, validation and verification data concerning specific case studies, types of soil, plant types, leaf area, plant height, regarding the Physical Twin, or technological use cases. Critical data processing depends on specific plant characteristics, experimental setups, dedicated sensors, ML and DL applicable methods and big data analytics software.

One of the significant benefits of utilizing the DT approach is its ability to accurately depict objects or entities that change over time, owing to the incorporation of dynamic and evolving data. Employing a digital twin makes it possible to extend the application of a validated model across timescales during which the object and its behavior undergo significant variations. A validated model may offer an instant representation of an object’s behavior at a specific moment, thus highlighting the importance of associating the DT with a corresponding physical counterpart and the significance of choosing a model according to the specific implementation.

Without a physical twin, the DT would be considered nothing more than a model. Furthermore, the availability of a comprehensive and up-to-date dataset plays a crucial role in accurately capturing the object’s changes over time and ensuring the model is adjusted and updated to reflect the current system conditions. Any model approximating the physical twinned can be used for DTs that utilize physics-based models encompassing all relevant processes that can influence the measured quantities. In an ideal scenario with instantaneous computing and flawless accuracy, these models would accurately simulate and update the mechanical and thermal processes involved in real-time operations. For instance, a DT of a machine tool could simulate the milling of metal, considering factors like temperature and part shape, and dynamically update information about tool wear based on real-time measurements. This enables proactive and effective industrial plant facilities maintenance, enhancing overall efficiency [62,63]. Similarly, to an industrial plant in agriculture, a digital model could represent the growth of a plant regarding its physical environmental conditions that are changing over time.

According to researchers, a categorization standard centered around the degree of data fusion and integration between the PO and its virtual representation can be used as an indicator to define and delimit the scope of the proposed concept. Given that this term was initially created for the manufacturing industry, it serves as a proxy for evaluating agricultural DTs’ development, convergences, and divergences compared to contemporary definitions. Thus, three integration levels represent the digital twin’s field. First is the Digital Model (DM), second is the Digital Shadow (DS), and third is the DT. The DM is a depiction of the entity in a digital format without the use of an automated data transfer system. It is equivalent to a Digital Twin prototype and represents the lowest possible integration level. The DS refers to a digital representation that enables the automatic exchange of information from the physical entity to its virtual object, reflecting changes in the entity’s state. This one-way information flow is like a DT model, which in contrast, involves an automated bi-directional flow of digital information. Like the DS, the DT has a virtual representation whose alterations are reflected in the physical entity’s state. However, what sets the DT apart is its capacity to impact the condition of the physical entity as well, although the specific means of influence depend on the type of entity and its context [63].

In summary, a DM represents a physical system in a digital format but lacks automated data exchange. A DS involves a one-way automated data flow reaching the digital counterpart originating from the physical system. Conversely, a DT involves seamless and fully integrated automated communication between the digital counterpart of the physical system and the physical system itself.

Defining the objectives of the modelling process is the initial stage in creating a model for an integrated process. Recognizing that a model is merely a reality approximation is essential, as diverse objectives can provide different levels of complexity in representing the actual procedure. For this reason, it is crucial to identify the goal of the model before the build [64]. Therefore, complex systems like an agriculture model should be first designed and objectively defined so their DT model is accurate. Spreadsheet software such as Microsoft Excel is commonly employed for process calculations and analyses. This is due to its widespread availability and familiarity among scientists, engineers, and other professionals. In contrast, the concept of DTs is somehow new and still emerging. The spreadsheet’s cells allow users to enter data, make computations, and produce results while they can plot data from them in several graphs. Software programs like process simulators allow users to describe and analyze integrated processes. The purpose of these tools is to simulate continuous processes and their transient behavior. The scientific literature concerning process simulation tools is continuously growing, encompassing various areas such as techno-economic analyses, economic feasibility studies, structural process enhancements, and the exploration of alternative processes to existing ones [64]. Given that a DT model is much more than a simulation tool, by nature, never being an exact representation of reality. Still, really an approximation of it, and different objectives may produce various modelling abstractions with varying levels of detail concerning the actual process. It is crucial to identify the goal of the model before we develop it.

Since cloud computing has many benefits, it is the most practical method for introducing DT services. Due to its ability to provide on-demand services, computational resources, and ubiquitous network access, the implementation of the next-generation information technology architecture is highly compatible with DT environments. In such environments, data owners generate data from assets, transmit it to cloud servers, simulate the DT in a virtual environment, and subsequently share the insights and findings of the simulation with the data owner. Users can request access to the data at any time [59]. In constructing complex DT production systems, like industrial or agricultural systems, “machine learning” and “simulation tools” are two complementary technologies. Creating stochastic simulations for complicated systems takes time and effort. However, these simulations are essential for DT construction because they have two crucial capabilities. The first is “uncertainty modelling” and the second is “explainable analytics”. Focusing on how ML and reinforcement learning (RL) improve simulation, many issues arise in digital twin use cases. The rapid advancement of IoT, edge computing, and cloud computing technologies has significantly accelerated the progress of digital twins. It is widely believed that digital twins will soon become a vital catalyst for digital transformation, as evidenced by the common theme present in various research studies [65]. The primary hurdle lies in establishing a secure, reliable method to share simulation and real-time data. Additionally, there is a need for a robust and privacy-centric authentication system that leverages the benefits of blockchain technology to address the security prerequisites mentioned earlier.

### 3.2. Notable Components of DT Implementations

In the realm of digital twins, components denote the distinct constituents or elements that constitute a digital twin system. A representative ensemble follows.

#### 3.2.1. Sensors and Actuators 

DTs embody sensors and actuators as parts of great significance that offer valuable data and information about physical phenomena and enable the replication of sensory capabilities. In today’s interconnected world, an enormous number of sensors and actuators are deployed across various domains, ranging from consumer products to industrial settings such as smart factories and smartwatches. These sensors capture critical information related to temperature, humidity, velocity, pressure, chemical components, and material composition, forming the foundation for digital twin technology [47,66]. By equipping real twins (physical entities) with sensors, DTs can replicate and harness the senses of sight, temperature, sound, humidity, and touch through appropriate actuators. For example, in the agricultural domain, a DT of a plant could leverage sensors to create predictive maintenance services or simulate and optimize the production process. This allows for the generation of actionable insights and the enhancement of operational efficiency. DTs rely on the integration of actuator and sensor technologies to gather continuous and accurate data. The representation of a physical asset in digital forms, like a sensor-equipped machine, relies on the synchronized operation of virtual sensors and actuators. These software-based counterparts replicate the functionalities of physical sensors and actuators, aggregating data from multiple sources. Integrating sensors and actuators within the digital twin framework enables real-time monitoring, data-driven analytics, and the ability to simulate various scenarios for optimization purposes. By bridging the physical and digital realms, digital twins empower industries and sectors to achieve enhanced performance, increased efficiency, and cost savings.

#### 3.2.2. Identification of Entities

In the context of DTs, each twin is assigned a distinct identifier that facilitates seamless communication with its physical counterpart. This unique identifier encapsulates vital information about the physical object’s structure, capabilities, and technical specifications, as well as its expected performance under specific conditions. When creating a DT, it is crucial to consider the unique circumstances of the object to ensure an accurate representation [67]. Even if two physical items are identical, they may exhibit different behaviors when subjected to varying conditions. Hence, each digital twin is assigned its unique identifier, reflecting the individual characteristics and attributes of the corresponding physical object. By leveraging the DT’s unique identifier, we can accurately simulate and analyze its properties and behaviours.

Furthermore, the expansive network of interconnected devices, sensors, and actuators highlights the need for real-time collection and sharing of identified contextual data with digital twins. This data encompasses various parameters such as location, environmental conditions, user preferences, and behavioral patterns. By integrating this uniquely identified contextual data into the DT model, a deep understanding of the user’s context and needs can be achieved. Adding to that notion, a Geographic Information System (GIS) as a versatile system generates, controls, examines, and maps various data types and links IoT data to a map. Thus, GIS combines location information, referred to as spatial data, with descriptive details of a DT, forming the basis for mapping and analysis utilized across multiple industries and scientific fields. With GIS, users gain insights into patterns, relationships, and the geographical context of their data, as in [68].

#### 3.2.3. Visualization and Representation 

Visualization and representation play a vital role in DTs, offering various forms of virtual representation depending on the specific application. DTs can leverage 3D visualizations to facilitate collaborative design or planning [69]. Depending on the context, these connections can manifest in various forms, such as a 3D representation, hologram, humanoid social robot, or purely as software components without physical attendance. Consequently, DT commonly refers to the digital depiction of a real-world object, system, or entity’s physical twin that faithfully replicates a unique item, process, person, organization, or even a legendary narrative. Composite digital twins are formed when multiple digital twins merge. These DTs, as reflections of the actual world, share similar characteristics, necessitating the modelling of digital processes and objects with composition, flexibility, sensing, collaboration, and simulation capabilities. Furthermore, predictions suggest that future DTs will evolve toward a holistic virtual world [70].

#### 3.2.4. Devices 

Devices that continuously capture data concerning physiological and environmental factors information can serve DTs to gain a comprehensive understanding of the real twin’s well-being and lifestyle. Such wearable devices, for example, smartwatches, fitness trackers, and other health monitoring devices have enabled individuals to collect and track various physiological parameters effortlessly. The abundance of such data presents a unique opportunity for digital twins to enhance their support for their real twins more efficiently [71]. By integrating the data from wearables into the DT model, real-time insights and personalized recommendations can be generated as feedback or alerts based on specific needs and goals. Furthermore, the continuous monitoring and analysis of wearable data by DTs enable proactive interventions whilst identifying patterns or deviations from normal physiological parameters and providing early warnings or interventions to address potential health risks or optimize performance. This proactive support can contribute to preventive healthcare, early detection of health issues, and overall well-being management.

In summary, the DT is an advanced data management system that integrates 3D animation, modelling, IoT, AI, and other technologies such as ML. It enables the seamless connection and management of physical and virtual models within a digital environment [72]. As human civilization grows, agriculture remains crucial for our progress and well-being. Intelligent DT systems can analyze and digitize agricultural areas, enabling macro and micro-level visual processes. These systems can provide intelligent recommendations and suggestions that can be implemented to achieve multiple benefits in agriculture.

## 4. Reviewing DT Applications in Agriculture and Farming Domain

Regarding the methodology, we opted for our research. Initially, an extended search for literature and bibliography referring to DT in agriculture was conducted in Google Scholar, Scopus, and Web of Science. We used the query item “digital twin” incorporating the logical operation of conjunction with the concepts of “agriculture” including concept association as “crop”, “farm”, “aquaculture”, “animal” and “smart farming” to step on cases of DT in subfields of agriculture. Queries as such provided 144 relevant papers from Google Scholar, 153 papers from Web of Science, and 135 papers from Scopus.

Secondly, we narrowed down all references to 105 papers by omitting duplicates, research papers with almost identical titles, with insufficient or less relevant content, and following our research individual issues related to the main criteria and title as initially stated in the introduction, i.e., crop modelling, precision agriculture, and predictive maintenance in smart farming via the DT concept.

Diving deeper into the selection criteria, we used keywords such as plants, smart agriculture, agriculture 4.0, IoT, aquaculture, food, horticulture, urban farming, DT models, 3D simulation, AR, VR, and Blockchain ledgers. Totally 89 papers, including those for initially referenced material, were chosen as being deemed relevant to the context of our research.

Deriving from the central idea of our research based on our study’s concepts, eminently disclosed in the preceding paragraphs of the current Section 2, we finally selected a total number of 26 papers fulfilling the criteria to utilize our review as graphically viewed in Figure 8.

Being familiarized with the relevant literature, we achieved insights into the extent of DT penetration in agriculture, addressing methodologies, designs, data analysis techniques, technologies, and use cases utilized in application domains to gain a deeper understanding of the approaches employed by others regarding the DTs in agriculture. The main idea lines of our study concepts, though, as previously stated, targeting DTs, plants, 3D modelling, simulation, IoT, and ICT gave reason to review the relevant literature. The main concepts of our study are graphically displayed in Figure 9.

### 4.1. The First Categorization of Researched Literature, According to the Main Research Objectives

Based on the breakdown of the 26 papers concerning the DT application domains and target applications related to smart farming as a subset of smart agriculture or agriculture 4.0, highlighting the case studies, we gathered a set of related concepts in Table 2. If a relevant text was found, the “√” symbol is included in the cells; otherwise, if not clearly or at all stated, the “-” symbol is used.

Ιn a total of 26 reviewed articles, 3 were relevant case study conference papers, one was a published scientific book, 9 were journal articles, and 12 were review papers. To visually assist in the analysis of the presented information in Table 2 regarding the type of reviewed literature, Figure 10 presents these types in graphical mode.

Fourteen articles either being descriptive as in [9] or practically using a case study project as in [9,19,26,28,35,64,74,75,76,77,79,80,81,82] address the significance of a DT case study in agriculture. The concept of DTs is mainly addressed in all reviewed papers. In 16 papers though [19,26,27,28,64,73,82], the DT concept is analyzed and utilized in more detail, thus posing the significance of using DTs in smart farming and agriculture.

According to the reviewed literature summarized in Table 2, smart farming as an implementation of agriculture 4.0 [19,26,27,28,29,45,63,73,75,76,77,79,80,81,82,83,84,85,86,87,88] includes smart irrigation strategies by monitoring and controlling various farming stages [28], implementing data acquisition techniques [27] for automatically controlling actuator systems in agriculture or even in high-tech data-driven greenhouses [79]. Furthermore, via simulation, monitoring, controlling, coordinating, and executing farm operations at agricultural sites [87], DTs improve predictive control in precision irrigation [28,73,88], greenhouse horticulture, and organic vegetable and livestock farming. Smart farming enhances remote detection and monitoring of vegetation and crop stress in agriculture [86], developing DT stages and forecasting plant yield in greenhouses, vertical farms, or outdoor fields [76,77], or even in urban and indoor farming, of vertical agriculture utilizing hydroponics, aeroponics, aquaculture, and aquaponics. Smart farming approaches the smart agriculture [85] as well as the sustainable agriculture 4.0 [80] sector to address precision farming and management [75,76] via the adoption of digital technologies and techniques in agriculture [27,29,64], leading the way to digital representations of plants [9], inventory quality in agriculture, greenhouse production flow, and food safety and quality within the food supply chain [45]. Although the DTs remain a major issue in our research criteria, in papers [29,45,63,83,84,85], they are implied as such. In [9], authors propose a functional DT model; in [86], they are proposed as a significant future issue to address and study.

### 4.2. The Second Categorization of Researched Literature, Related to Specific Technical Research Aspects

Table 3 below gathers various aspects of the target applications related to sensors, IoT or other smart platforms, smart agriculture, and DT technologies and protocols implemented in the addressed literature for comparison and evaluation. If the topic of a corresponding cell has no relevant data provided in the literature “-” sign is displayed in the corresponding cell. If the topic is merely mentioned, a “√” sign is displayed in the corresponding cell.

DTs go along with monitoring systems to gather and analyze information. Sensors pose a critical role in smart agriculture by facilitating the collection of real-time data and offering valuable insights into a wide range of environmental and agricultural parameters.

Field or soil probes that measure air and ground temperature, humidity, soil moisture, and ambient light [28,45,73,81,82,88], added to CO_2_ sensors measuring relative CO_2_ concentration [80,81], and infrared (IR) thermometers provide data acquisition modules for automatically controlled actuator systems in agriculture approaching digital modelling to the food process [64] monitoring activities of livestock, optimization of crops, reducing emissions to air, soil, and water. Drone image cameras [88] and high-resolution photographic cameras seem ideal for AR and VR by constructing virtual natural low-polygon 3D plant models as proposed in [35,74]. Building Information Modeling (BIM) models of high-tech data-driven greenhouses are made feasible as in [79] and provide a VR presentation of the farm or field with the aid of a head-mounted display (HMD) with two handheld controllers serving as handy sensors for manual stimulus. Mini light detection and ranging (LiDAR) sensors and multispectral cameras following an Arduino single-board microcontrollers programmable platform or single-board computer Raspberry Pi-3 enhance IoT sensor nodes to acquire and transmit farm data to IoT gateways or edge devices, thus cultivating DT requirements for vertical farms, for the greenhouse production process and finally models in sustainable agriculture.

Remote detection and monitoring of vegetation and crop stress in agriculture seem feasible based on [86] utilizing LiDAR—stereo-photogrammetry technology, using multi-spectral imagery, passive microwave remote sensing, RFID schemes [9,45], active microwave remote sensing (RADAR) and sensors onboard UAVs and satellites [45,76,88]. Location determination in farming via geospatial position (GPS) plays a significant role in the remote detection and monitoring of vegetation and crop conditions in agriculture [88].

IoT deployments in smart farming are based on serial hardware communication protocols such as I2C serial bus [28] and UART that uses asynchronous serial communication with configurable speed and pulse width modulation (PWM) for driving actuators [77,81], in correlation with programmable logic controllers (PLC) for data recording [29], monitoring, and optimization aided occasionally by fuzzy interference systems (FIS) [73].

The decision-making and actions once undertaken by the farmers now have given way to data-driven agricultural decision-making, i.e., assisting them to undertake informed decisions related to planting, fertilization, pest control, harvesting, and overall farm management. Likewise, derived from the smart water management project (SWAMP) [28,73,75] a hands-on approach is being utilized to develop a precision irrigation platform based on IoT technology for smart water management, distributed in Brazil, Italy, and Spain.

### 4.3. The Third Categorization of Research Literature That Is Related to Specific Technical Research Aspects

DTs use software to create, operate, and interact with their physical counterparts. The software component of a DT enables the collection, processing, and analysis of data from various sources, including sensors, actuators, and other connected devices.

In [28,77], the high-level programming language Python is used for software development, scripting, and data analysis, serving the purposes of DT applications, implementing the Arduino IDE platform, and exploiting the Debian Buster OS to produce a plant simulation model. Likewise, RPi software for the Raspberry Pi platform is used with goal-oriented requirement language (GRL) for modelling [80]. We should point out the use of an open standard file format JavaScript Object Notation (JSON) which is used in [73] for data representation and communication between systems and is widely adopted and supported by various programming languages and platforms. Specific software packages are utilized for constructing 3D plant models, such as Unreal Engine 5, Reality Capture, Photoshop, Mesh Model Construction, Autodesk, Maya [74], Unreal Engine 5 Nanite technology, and Reality [35]. Other specific software packages serving, for example, the construction of a DT flowsheet model as in [64] found in the researched literature are the Aspen Plus and Aspen HYSYS from Aspen Technology, Inc. (Burlington, MA, USA), ChemCAD from Chemstations, Inc. (Houston, TX, USA), UniSim Design from Honeywell (Charlotte, NC, USA), ProSimPlus from ProSim SA (Labege, France) and PRO/II from AVEVA Group plc.

Simulation software was employed in this study [28] to generate a virtual setting for an irrigation system’s digital twin, incorporating a plant simulation model.

As previously mentioned, in [73], Siemens’s industrial plant simulation software was used, and in [74], the Virtual UCF Arboretum Application (V1.0) was developed with virtual plant datasets, plant inventories, VR headsets leading to an AR Holodeck by multiple captured images taken in 3D space and an AR perpetual garden App.

In [75], a linear model of plant growth is proposed utilizing a wheat multi-agent planning module close enough to the implied structure modelling simulation in [29], virtual models are used during the usage phase in [84], an earth system model in [78], a farm is represented in 3D in [81], the concept of the DT model via data-driven modelling is assessed [63]. In [76], the authors imply that a software package was developed containing an ontology editor, a DT editor, multi-agent planning module for creating a prototype of an intelligent plant DT system in Java whilst AR implementations to save BIM model files in Film box (.FBX) in [79] by using Unity game engine to provide 3D modelling software. Further authors in [9] confronted post-harvest models by discrete event simulation such as drying, adding to the simulation models for blending and flow DT models [82,83] that rely heavily on available real-time data a flow for continuous adaption and learning. Finally, DT modelling implies simulation, analysis, and prediction [88] that may assess the modelling and simulation of the fertility of seeds, fertilizers, pesticides, pollution challenges, soil agents (hydrological models, soil data), and crop agents

Table 4 summarises software, 3D, 3D modelling, simulation, and other research aspects addressed in our work regarding the reviewed literature. Suppose the topic of a corresponding cell has no relevant data provided in the literature, a “-” sign is displayed in the corresponding cell. If the topic is merely mentioned, a “√” sign is displayed in the corresponding cell.

On the other hand, AI, ML, and DL algorithms seem to be a major issue in [45,63,83,87,88], leading to Big Data analytics [82] to describe DT modelling that relies heavily on real-time available data flow. The immense amount of data is stored in databases such as Mongo DB [28,73], Draco, My-SQL [28,73], and SQLite [80,81], besides using Excel CSV files for retrieving height values [79] or generally mentioned Knowledge bases as in [75] leading to cloud big data manipulation schemes in [45,81,85]. Major cloud computing platforms such as Alibaba Cloud, Amazon web services, Google Cloud Platform, IBM, and Microsoft Azure are examples of cloud computing services provided by leading technology companies that serve the DT idea, as seen in [87].

Cloud IoT service broker agents come into place as a software component or intermediary that facilitates the interaction between cloud service consumers (users or organizations) and cloud service providers like FIWARΕ. In [28,73], FIWARΕ cloud applications are exploited, adding to IoT brοker and IoT agents to facilitate cloud management of context information. The procedure includes converting various communication protocols into a shared base protocol and incorporating intelligent functionality through the processing, analysing, and visualising of contextual data. FIWARE generally is used as an open-source platform to provide a standardized framework and a set of reusable components for building smart applications and services in the context of the IoT and Future Internet (FI) domains. It aims to simplify the development of innovative and interoperable solutions by offering a collection of open APIs, data models, and software tools that facilitate the development of smart applications and services.

A fuzzy inference system enhances adaptive and learning capabilities. It handles nonlinear and complex relationships between variables, as in [73], in addition to Siemens industrial plant simulation software for plant simulation model production with a weather station. In [67,79], the Microsoft Excel spreadsheet application is used instead of other dedicated software such as Geo-Soft-Core, a Geoscientific Software & Code Repository hosted at the archive DIGITAL.CSIC in [78] or Java ontology editor to elaborate a DT editor in [75,76].

Lastly, wireless network communication technologies are addressed in the research literature. Initially, the wired networking technology Ethernet is mentioned in [28,73,77], while the wireless networking technologies such as Wi-Fi are mentioned in [35,45,74,77,87], LoRaWAN in [45,73], mobile or cellular in [45,74,87] and generally mentioned as mandatory in [29,82]. Bluetooth, RFID, and NB-IoT are exploited in [85].

## 5. Discussion

A first attempt to comment on specific aspects of the reviewed literature has been done in Section 2. In total, 14 journal articles or conference papers, including a published book, propose case studies that support the idea of DT implementation in agriculture, contrary to the rest published review papers that suffice to merely describe the topic and record the achievements of the scientific community referring to it.

It is interesting to point out that three (3) relevant case study conference papers and a published book, added to nine (9) journal articles, seem to slightly overcome the amount of the 12 review papers in a total of 26 articles. The limited practical application of digital twin technologies is obvious, even stated in the literature by the authors.

Even though [9,86,87] are review papers, the authors either describe the necessity of a DT model as in [9] for the agriculture supply chain management domain, even addressing sensors such as level, flow, identification sensors, RFID or DNA tags, or highlight the need for future study on DT models as in [86] making use of remote detection and monitoring schemes, LiDAR sensors, passive microwave sensing devices, RADARs or even sensors onboard UAVs and satellites, for monitoring, controlling, coordinating, and executing farm operations at agricultural sites by IoT sensor nodes as in [87].

The significance of using Sensors as part of a dynamic IoT control-oriented mechanism as demanded in DT infrastructure should be acknowledged as in [35,73,74,77,79,80,81,87,88] where soil probes, drone image cameras, multispectral and photographic cameras, IR thermometers, mini-LiDAR sensors, a mesh of sensors measuring temperature, humidity, luminosity, and relative CO_2_ concentration, and air velocity are addressed as essential elements to materialize relevant IoT platforms accordingly.

In [28,73,75,88], the SWAMP IoT-based smart water management platform is mentioned. In [19,26], the IoT scheme is merely addressed, and in the journal [82], IoT is only mentioned as well in the review papers [63,85,87], where the IoT is stated as a promising concept. The fact that in [73] an OPC UA server is recommended for implementation, in [77] an Arduino single-board computer is used as well as the Raspberry Pi-3 platform as in [80,81] in a total of 14 articles that either describe the use of IoT platforms or vaguely present a case study project [9,19,26,28,35,64,74,75,76,77,79,80,81,82] means that the practical implementation of IoT platforms is quite limited for the time being.

The necessary software to operate such platforms is focused on Python [28,77], dedicated FIS, Json, FIWARE Cygnus connector, IoT Agent, OPC agent [73], RPi in [77,80], Arduino IDE n [77] relevant to the case studies previously mentioned. These make use of Ethernet communication technologies [28,73,77], LoRa [73], and Wi-Fi, [77] in addition to [35,74] that make use of Wi-Fi and mobile LTE [74] communication technologies that facilitate near real-time interaction with the environment.

The concept and definition of DT seem to be continuously evolving, but in general, it refers to a specific collection of digitally or virtually materialized models that represent a physical process or object. These models are used for simulation and future behavior forecasting, guiding the feedback loop from the Actuators and Control node to the Physical entity, as profoundly visualized in Figure 3, Figure 4 and Figure 5. The ongoing feedback loop between the physical farm and the DT is imperative to update the DT model to optimize and enhance performance and efficiency.

Several attempts have been made in this direction. In the reviews [9,29,63,83,86], the concept of the DT model is proposed merely as an evolving concept without an implemented case study. Simulation models [83], referring to data-driven modelling, discrete event simulation blending and flow models [9], and structure modelling simulation and future multi-domain radiative transfer models (e.g., SCOPE) with dynamic crop growth models for agroecosystems DTs [86] are described merely as proposals giving way to future practical implementations. On the other hand, in [28], simulation software utilizing a simple plant simulation model is employed to create a virtual environment for an irrigation system’s digital twin. In [73], the Siemens industrial plant simulation software for the data model is used, a model of the Earth system is mentioned in [78], a 3D representation of a farm in [81], while in [82] it continuous adaption and learning procedure of the DT modelling is mentioned as a need that relies heavily on available data. A more concise proposal to the origin of the holistic DT modelling and simulation concept appears in [76], where a prototype of an intelligent plant DT system in Java claimed to be developed, including an ontology editor, a digital twin editor, and a multi-agent planning module. In [88], simulation, analysis, and prediction are addressed by making progress in implementing modelling and simulation of the fertility of seeds, fertilizer, pesticides, and pollution challenges, making efforts to develop plant modelling with soil agents (hydrological models, soil data), crop agents and predictive control models, where the adjustment of control processes such as heating and ventilation is automated based on short-term temperature predictions, ensuring efficient regulation, as an IoT derivative scheme.

A flowsheet model is proposed taking advantage of the widespread spreadsheet applications of Microsoft Excel in [64] to tamper with simulation software such as Aspen Plus and Aspen HYSYS from Aspen Technology, Inc. (Burlington, MA, USA), ChemCAD from Chemstations, Inc. (Houston, TX, USA), UniSim Design from Honeywell (Charlotte, NC, USA), ProSimPlus from ProSim SA (Labege, France) and PRO/II from AVEVA Group plc, merely to implement a digital modelling approach to food processing.

Spreadsheet applications of Microsoft Excel to retrieve height values of plants in high-tech data-driven greenhouses are used in [79], a fact that proves that process calculations and analyses may be conducted using popular spreadsheet software like Microsoft Excel due to its wide accessibility and familiarity among scientists, engineers, and professionals from various fields.

Enhanced 3D modelling software derived from the game industry, i.e., Unity game engine dictates BIM models in Film box (.FBX) format to aid realistic VR/AR representations of greenhouses, providing a promising realistic overview of a Smart Agriculture implementation. Virtual representation is accomplished via Unreal Engine 5, Reality Capture, Photoshop, Autodesk, and Maya software platforms to construct Mesh models of Plant Datasets for Virtual UCF Arboretum Applications, plant inventories, and AR Perpetual Garden App. as in [74]. These plant datasets are derived from multiple captured images in 3D space and incorporate VR headsets aided by GIS and AR Holodeck software. Additionally, Unreal Engine 5 Nanite technology and Reality are used for 3D plant models offering a linear plant growth model [35] to descriptively suffice the wheat multi-agent planning module. Using multiple captured images taken in 3D space to support the AR Perpetual Garden App, they produce botanically correct plant models for use in museums and for educational purposes. A promising perspective, if by any chance associated with the essential elements of the holistic DT scheme, could be capturing real-time data on various environmental and operational factors via communication technologies leading to a dynamic 3D model of a farm entity.

Besides the 3D representation of DT models, IoT Dashboards [87], such as Grafana in particular [28,73] for real-time data presentation and analysis or even GUI prototypes [81], are proposed, although used on a small scale for facilitating near real-time interaction of DTs with the environment. To reliably transfer and manage data, Blockchain is used, even on a smaller scale, as in [45,87], while Big Data Analytics, as in [82], AI and ML methods that enable data analysis, pattern recognition, and decision-making based on large datasets, are addressed in [45,63,82,83,86,88]. Cloud-based data analytics pose a major issue for the advent of smart agriculture as claimed in [45,85,87], not to omit the significance of databases such Mongo DB, Draco, My-SQL as used in [28,73] or SQLite in [80], and Cloud computing service providers such as Alibaba Cloud, Amazon web services, Microsoft Azure and Google Cloud platform [87].

Various factors, with technical hurdles being the primary challenge, seem to be stepping behind to allow even further the adoption of the DT technologies. It is essential to meticulously design and clearly define intricate agricultural systems to guarantee the accuracy of their corresponding DT models.

## 6. Future Directions

Once addressing a DT as a tight combination of an object model, in conjunction with expansive sets of data that directly follow the object that may evolve and continually modify and adapt the model based on the available data, it is made clear that future research should be realized to overcome the current profound lack of reference models and case studies accordingly. That could serve as a direct guidance for DT research and development.

To broaden the aspect of Agriculture 4.0 that needs DTs and overcome the limitations of mere telemetry narrowed down by IoT schemes implemented till now, further research in implementations using DT modelling, 3D visualization, and simulation schemes targeting virtual nature applications aided by AR, VR, and GIS information systems should be realized.

Further research into the holistic model perception of plant DTs to overcome the focus merely on the phenotype of plants could vastly support DT research. We can pursue enhancing the accuracy of depicting real-world farms, plants, and crops in the digital realm by combining non-georeferenced visualizations, like gaming engines, with geospatially referenced 3D geo visualizations.

Implementing additional case studies of digital twins that adhere to the mentioned principles can address challenges related to reliable aggregating and handling of data from diverse sources, integrating multiple streams of information, and employing advanced computational methods for data processing and analysis, which could enhance the understanding of the subject and aid further studies.

Further decommissioning of data in the future resulting from search—Table 1, Table 2 and Table 3 might be promising. The diverse and abundant potential advantages of DTs in agriculture create anticipation for the future evolution of this technology and its profound impact on the farming sector.

We aspire for our work to inspire fellow researchers to explore the application of DTs in agriculture even further.

## Figures and Tables

**Figure 1 sensors-23-07128-f001:**
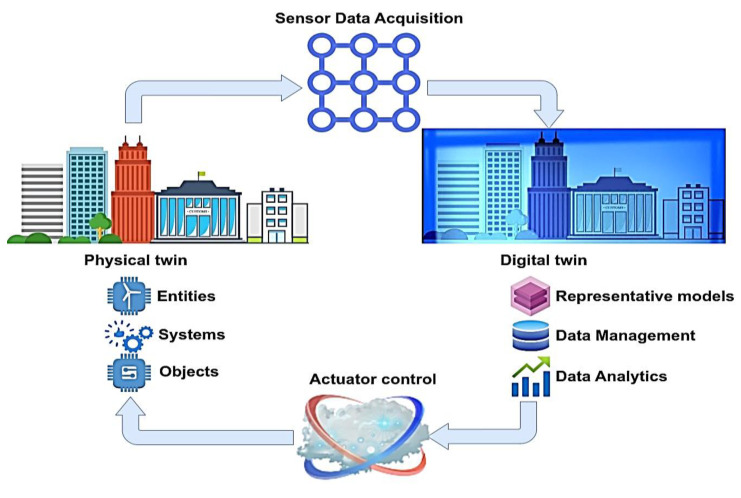
The generic representation of a DT scheme’s dynamic integration of physical and digital domains.

**Figure 2 sensors-23-07128-f002:**
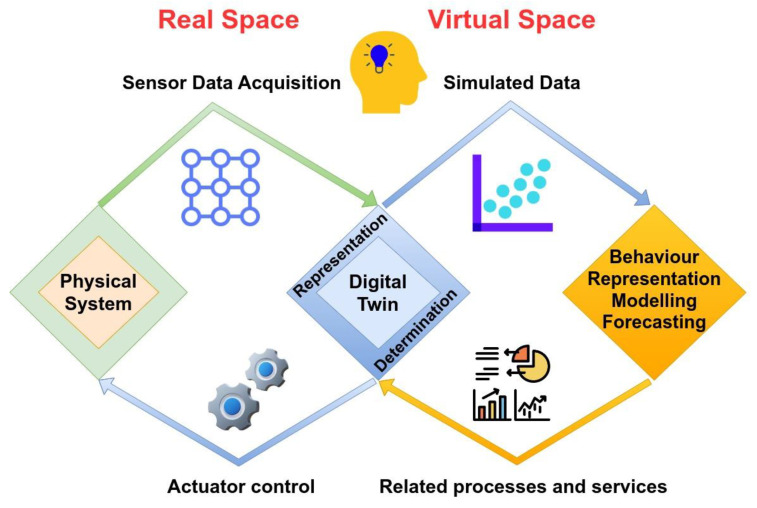
A simplified view of the data interaction between the physical entity and its digital representation in a DT.

**Figure 3 sensors-23-07128-f003:**
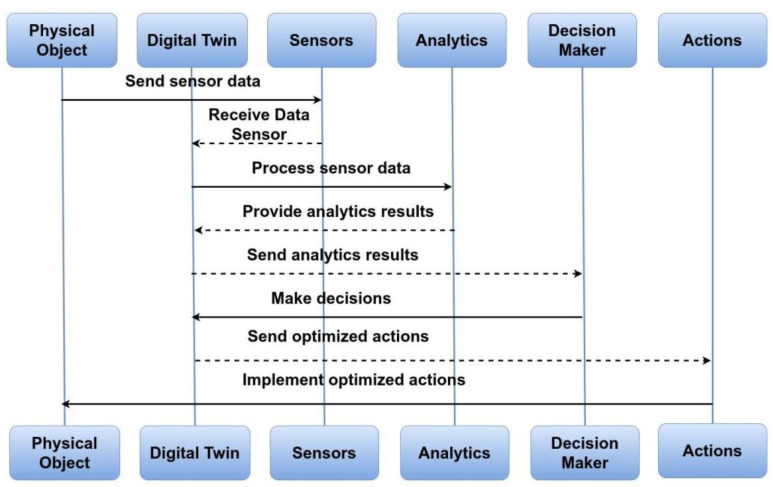
Interactions sequence chart between Digital Twin components.

**Figure 4 sensors-23-07128-f004:**
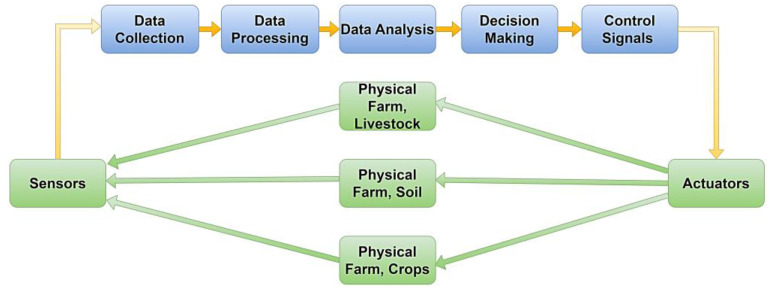
A schematic presentation of a DT concept scheme integration in Agriculture.

**Figure 5 sensors-23-07128-f005:**
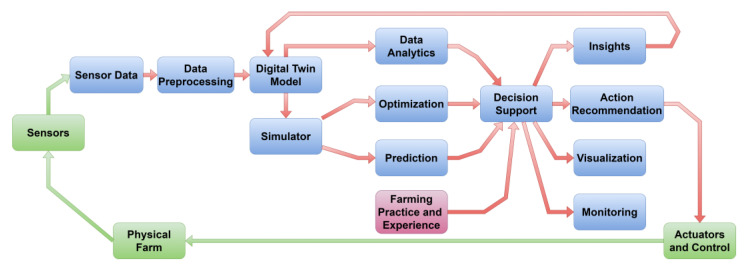
The Architectural Framework for Implementing DTs in Agriculture.

**Figure 6 sensors-23-07128-f006:**
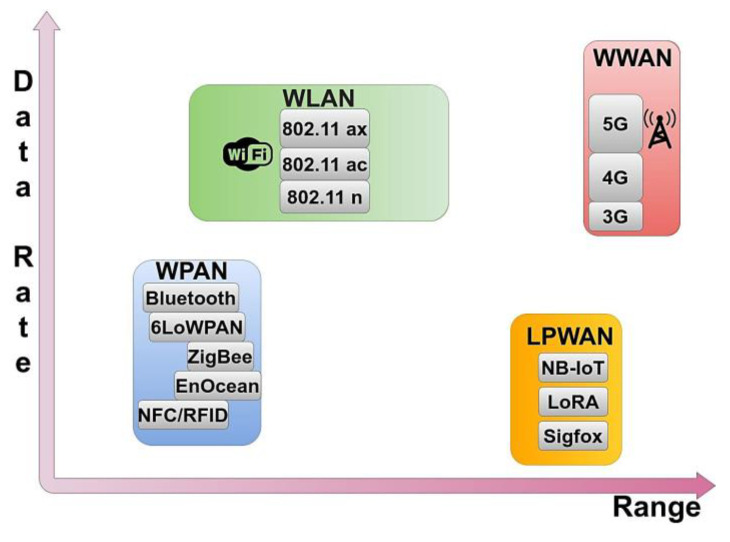
Data Rate vs. Range.

**Figure 7 sensors-23-07128-f007:**
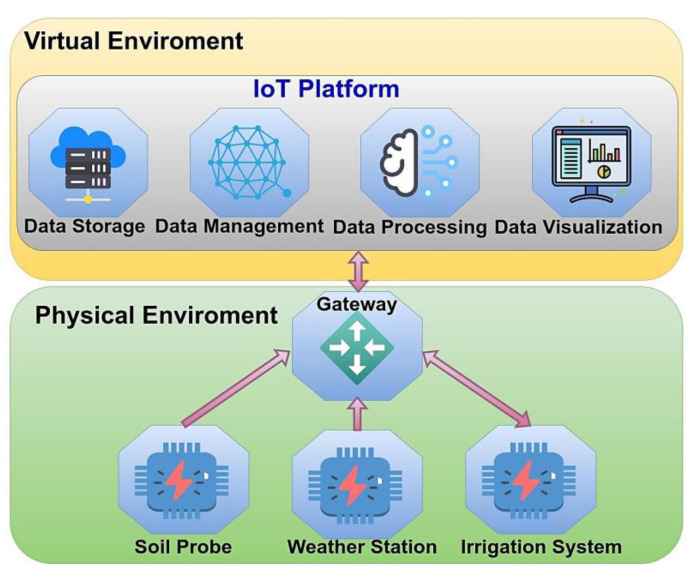
Connection entities over an IoT platform overview.

**Figure 8 sensors-23-07128-f008:**
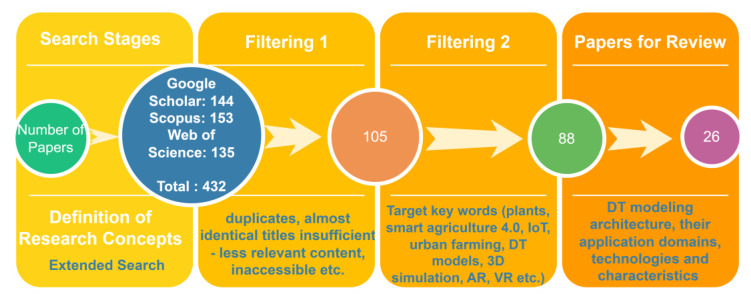
Schematic presentation of the paper selection procedure for reviewing.

**Figure 9 sensors-23-07128-f009:**
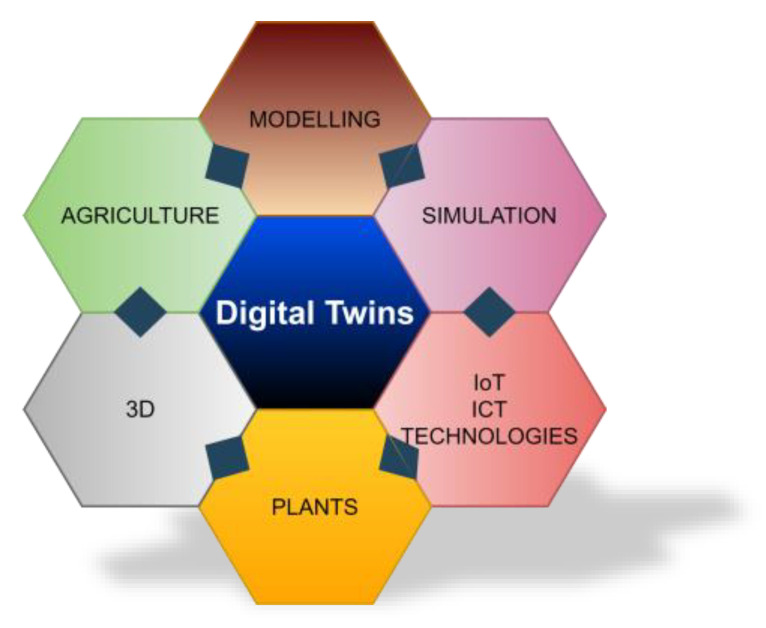
Crucial concepts.

**Figure 10 sensors-23-07128-f010:**
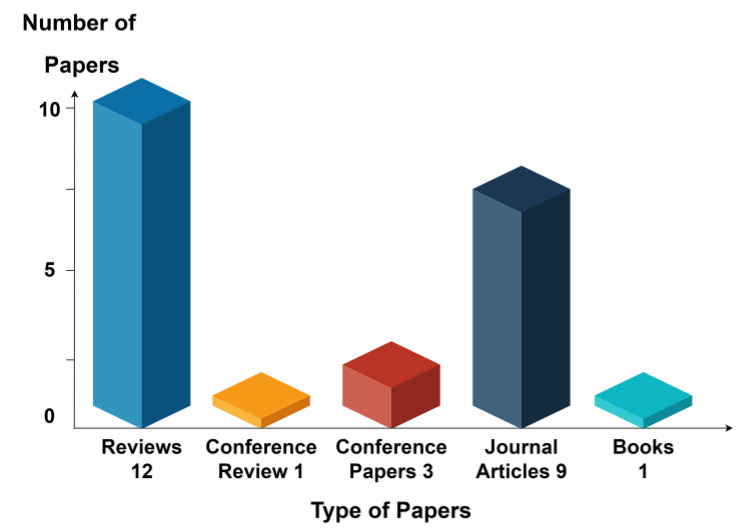
Review paper type distribution.

**Table 1 sensors-23-07128-t001:** Main features of Wireless Network technologies.

Wireless Networks	Range of Coverage	Frequency Bands	The Data Rate for Downlink and Uplink	Maximum Payload Length	Security	Battery Life
LoRa	Urban: 1–5 km Rural: 10–40 km	for Asia: 433 MHz for Europe: 868 MHz for America: 915 MHz	0.3–50 kbps	243 bytes	Medium (AES-128)	10 years
NB-IoT	Urban: 1 km Rural: 10 km	EU: 800, 900, 1800 MHz Middle East-North Africa: 800, 900 MHz North America: 600, 700, 850, 1700 MHz Asian Pacific: 700, 800, 850, 900, 1800, 2000 MHz	0.5–200 kbps	1600 bytes	High (LTE Security)	10 years
Bluetooth	Bluetooth 4.x to 100 m, Bluetooth 5.x to 200 m	2.4 GHz	Bluetooth 4.x 1 MB/s, Bluetooth 5.x 2 MB/s	Bluetooth 4.x 31 bytes, Bluetooth 5.x 255 bytes.	Low (AES-128)	some weeks
RFID	LF: 1 m HF: 10 m UHF: 100 m	125–135 kHz (LF) 13.56 MHz (HF) 860–960 MHz (UHF)	LF: 100–300 bps HF: 1 kbps UHF: 40–640 kbps	LF: 64 bits HF: 128 bits UHF: 256 bits	Low (Encryption-Wake Algorithm)	Passive: 3–5 years
WIFI	max 300 m	2.4 GHz 5 GHz 6 GHz (802.11 ax)	802.11/n:600 Mbps /ac: 1.3 Gbps /ax: 9.6 Gbp/s	no limit	High (AES, WPA2/WPA3)	power supply

**Table 2 sensors-23-07128-t002:** The main research objectives and content topics of eligible referenced articles are included in the review.

Citation No.	Type of Paper	Case Study	Smart Farming	DT Case	Application Domains	Target Applications
[28]	Journal Article	√	√	√	Agriculture, farming	Precision irrigation
[73]	Journal Article	√	√	√	Agriculture, farming	Precision irrigation for water saving
[19]	Journal Article	√	√	√	Arable, dairy, and farming livestock farming	Greenhouse horticulture, organic vegetable farming
[26]	Journal Article	√	√	√	Arable farming, dairy farming, greenhouse horticulture	Organic vegetable and livestock farming, smart farming
[74]	Case Study on Conference Paper	√	-	√	Virtual nature applications, the digital twin of natural environments	Museums, arboretums, field trip experiences, botanical gardens
[35]	Conference Paper	√	-	√	Art realistic and botanically correct plant models	Game design, environmental art, GIS, and computer science
[75]	Conference paper	√	√	√	Precision farming, management	Construction and implementation of plant DT
[76]	Book	√	√	√	Precision farming, management	DT development stages and forecasting plant yield
[77]	Journal Article	√	√	-	Automated irrigation via soil water monitoring	Greenhouses, vertical farms, or outdoor fields
[78]	Systematic Review	-	-	√	Monitoring, modelling, and forecasting natural processes	Geoscientific software & code repository
[79]	Journal Article	√	√	√	Smart farming	High-tech data-driven greenhouses
[80]	Conference Paper	√	√	√	Sustainable agriculture 4.0, vertical arming	DT cultivating model in sustainable agriculture
[81]	Journal Article	√	√	√	Vertical farming-greenhouses and bioeconomy	DT requirements for vertical farms
[82]	Journal Article	√	√	√	DT of greenhouse production flow, energy-efficient	DT for the greenhouse production process (WP4)
[64]	Journal Article	√	-	√	DTs in agriculture	A digital modelling approach to the food process
[27]	Review Paper	-	√	√	DT adoption in agriculture	Data acquisition for automatically controlled actuator systems in agriculture
[29]	Review Paper	-	√	-	Digital technologies and techniques in agricultural contexts—food post-harvest processing in the agricultural field	A general framework of digital twins in soil, irrigation, robotics, farm machinery
[9]	Review Paper	Descriptive	-	Model	Digital representation of grain and inventory quality in agriculture	Agriculture supply chain management
[83]	Review Paper	-	√	-	Controlled farming environment	Monitoring activities of livestock, optimization of crops, reduction of emissions to air, soil, and water
[63]	Review Paper	-	√	-	Agriculture, farming, crops, livestock	New farming methods supported by the DT
[84]	Review Paper	-	√	-	Greenhouse horticulture, indoor farming smart agriculture	IoT Data-driven food production.
[85]	Review Paper	-	√	-	Industry 4.0 approaches to the agricultural sector	Virtualization of an agro-food supply chain
[45]	Review Paper	-	√	-	Food safety and quality, supply chain	Authenticity and traceability of food supply in the agricultural production process
[86]	Review Article	-	√	future study	Sustainable and precision agriculture	Remote detection and monitoring of vegetation and crop stress in agriculture
[87]	Review Conference Paper	-	√	√	Urban farming, vertical farming, indoor farming, hydroponics, aeroponics, aquaculture, and aquaponics.	Monitor, control, coordinate, and execute farm operations at agricultural sites
[88]	Review Paper	-	√	√	DTs applied to precision agriculture.	Predictive control, for improving soil quality

**Table 3 sensors-23-07128-t003:** Various aspects of the target applications concerning sensors, IoT, or other smart platforms relevant to smart agriculture and DT technologies and protocols.

Citation No.	Case Study	Type of Paper	DT Case	Target Applications	Sensors	IoT/Platforms	Deployments/ Technologies- Protocols
[28]	√	Journal Article	√	Precision irrigation	Field probes measure air and soil temperature, humidity, soil moisture, ambient light, geospatial position, (Venus GPS)	SWAMP-IoT smart water management, smartphone application, real-time IoT platform communication.	I2C serial bus, RPi-3 module
[73]	√	Journal Article	√	Precision irrigation for water saving	Soil probe	SWAMP, OPC UA server	Fuzzy Interference System, Programmable Logic Controller (PLC)
[19]	√	Journal Article	√	Greenhouse horticulture, organic vegetable farming	-	IoT addressed	-
[26]	√	Journal Article	√	Organic vegetable and livestock farming, smart farming	-	IoT addressed	-
[74]	√	Case Study on Conference Paper	√	Museums, arboretums, field trip experiences, botanical gardens	Drone image cameras, Photographic cameras,	-	-
[35]	√	Conference Paper	√	Game design, environmental art, GIS, and computer science	Photographic cameras,	-	Virtual nature construction low-polygon 3D plant models ideal for augmented reality (AR) and virtual reality (VR)
[75]	√	Conference paper	√	Construction-implementation of plant DT	-	SWAMP	An intelligent digital twin of plant, IDT
[76]	√	Book	√	DT development stages and forecasting plant yield			Proposal for an intelligent digital twin of plant, IDT
[77]	√	Journal Article	-	Greenhouses, vertical farms, or outdoor fields	Infrared (IR) thermometers, mini-LiDAR sensors, multispectral cameras	Arduino board, single-board computer Raspberry Pi-3	I2C, serial peripheral interface (SPI), UART
[79]	√	Journal Article	√	High-tech data-driven greenhouses	Handheld controllers, with a button and joystick interaction functionality	-	BIM model, A Meta Quest2 head-mounted display (HMD) with 2 handheld controllers/VR
[80]	√	Conference Paper	√	DT cultivating model in sustainable agriculture	Mesh of sensors of temperature, humidity, luminosity, and relative CO_2_ concentration	Raspberry Pi	-
[81]	√	Journal Article	√	DT requirements for vertical farms	Temperature, humidity, luminosity, and relative CO_2_, Data Acquisition module	Raspberry Pi	Digital, PWM, I2C, SPI, Serial, network attached storage, Sampling 42sec
[82]	√	Journal Article	√	DT for the greenhouse production process (WP4)	Mentioned traditional sensor data	mentioned	-
[64]	√	Journal Article	√	A digital modelling approach to the food process	-	-	Automatically controlling system actuators
[27]	-	Review Paper	√	Data acquisition for automatically controlled actuator systems in agriculture	-	-	-
[29]	-	Review Paper	-	A general framework of digital twins in soil, irrigation, robotics, farm machinery	-	-	Data recording, artificial
[9]	Descriptive	Review Paper	Model	Agriculture supply chain management	Bin level, flow, and identification sensors, RFID-DNA tags	-	-
[83]	-	Review Paper	-	Monitoring activities of livestock, optimization of crops, reduction of emissions to air, soil, and water	Report	-	-
[63]	-	Review Paper	-	New farming methods supported by the DT	Monitoring physical entity crops state, resource optimization, and cultivation support	IoT stated	-
[84]	-	Review Paper	-	IoT Data-driven food production.	-	Climate control, energy, and lighting management	Monitoring, optimization for controlling and autonomy
[85]	-	Review Paper	-	Virtualization of an agro-food supply chain	-	IoT concepts	-
[45]	-	Review Paper	-	Authenticity and traceability of food supply in the agricultural production process	Data acquisition—temperature, humidity, soil conditions, location, RFID	-	Satellites Robotics technology assisted with AI, ML, and deep-learning techniques
[86]	-	Review Article	Future study	Remote detection and monitoring of vegetation and crop stress in agriculture	LiDAR-light detection and ranging—stereo-photogrammetry using multi-spectral imagery. Passive microwave remote sensing. Active microwave remote sensing (RADAR). sensors onboard UAVs and satellites	-	-
[87]	-	Review-Assessment	√	Monitor, control, coordinate, and execute farm operations at agricultural sites	IoT sensor nodes	IoT sensor nodes	IoT sensor nodes acquire and transmit farm data to IoT gateways or edge devices
[88]	-	Conference paper	√	Predictive control, for improving soil quality	Soil probes sensors temperature, relative humidity (RH), CO_2_ concentration, air velocity, and light level sensors. Drones	IoT, SWAMP	Programmable logic controllers (PLCs) in the irrigation system, equipment and machines

**Table 4 sensors-23-07128-t004:** Various aspects of the target applications concerning communication technologies, IoT, and cloud platforms for data and simulation processes.

Citation No.	Communication Technologies	Real-Time Data, Visualization, Analytics	IoT Cloud Services	Data Bases	Software	Simulation Software	3D, Modelling, AR-VR
[28]	Ethernet	Grafana, Real-Time Data	IoT Broker, FIWARΕ, IoT agent	Mongo DB, Draco, My-SQL,	Python	Simulation software to generate a virtual environment for a DT of an irrigation system	Plant simulation model
[73]	LoRa, Ethernet	Grafana, Real-Time Data	FIWARE IoT Agent	My SQL, MongoDB	Fuzzy Inference System (FIS), Json, FIWARE Cygnus connector, IoT Agent, OPC UA agent	Siemens Ind. plant simulation software for the Data model and weather station,	-
[19]	-	-	√	-	-	-	Conceptual DT modelling.
[74]	Wi-Fi. Mobile	AR/VR Software	√	-	Unreal Engine 5, Reality Capture, Photoshop,—Mesh Model Construction, Autodesk, Maya	Virtual UCF Arboretum Application, ESRI GIS, Plant Datasets, Plant Inventories and Density, VR Headset, AR Holodeck	Multiple captured images were taken in 3D space, AR Perpetual Garden App
[35]	Wi-Fi	-	√	-	Unreal Engine 5 Nanite technology and Reality for 3D plant models	-	Multiple captured images were taken in 3D space/AR Perpetual Garden App
[75]	-	-	-	Knowledge Base	Java	A linear model of plant growth	A descriptively wheat multi-agent planning module
[76]	-	-	-	-	Java ontology Editor, digital twin editor, the multi-agent planning module	The software package developed claimed an ontology editor, a digital twin editor, a multi-agent planning module	Prototype of an intelligent plant DT system in Java
[77]	Wi-Fi, Ethernet	-	-	-	Debian Buster OS, Python, Arduino IDE	-	-
[78]	-	-	-	-	Geo-Soft-Core, a Geoscientific Software & Code Repository, hosted at the archive DIGITAL.CSIC	-	Model of Earth system
[79]	-	-	-	Height value retrieved from CSV files	Spreadsheet applications—Microsoft Excel,	3D modelling software. BIM model in Film box (.FBX) Unity game engine	AR
[80]	-	-	Yes	SQLite	RPi software, Goal-oriented Requirement Language (GRL) for modelling	-	GRL model
[81]	-	GUI prototype		SQLite	-	-	3D representation of farm
[82]	mentioned	Mentioned-Industrial Data Management System multilayer approach with Developed IoT models	Big Data only Mentioned, cloud-based enterprise	-	AI, Big Data analytics	Mentioned	DT modelling relies heavily on available data and a continuous flow of real-time for continuous adaption and learning
[64]	-	-	-	-	SuperPro Designer, Spreadsheet applications-Microsoft Excel,	Aspen Plus/HYSYS ChemCAD (Chemstations, Inc.), UniSim Design (Honeywell), ProSim Plus (ProSim SA), PRO/II (AVEVA Group plc)	Flowsheet model
[29]	IoT, wireless technologies	Analysis, prediction	-	-	-	-	Structure modelling simulation
[9]	-	-	-	-	-	-	Post-harvest models Discrete event simulation Drying, Blending and Flow models
[83]	-	IoT, wireless technologies	-	-	ML	-	Simulation models
[63]	-	-	-	-	ML and DL algorithms	-	Concept of the DT Model, Data-driven modelling
[84]	-	-	-	-	-	-	Virtual models during the usage phase
[85]	Bluetooth, RFID, NB-IoT	-	Farm activities connected to the cloud	-	Big Data GPS,	-	-
[45]	Wi-Fi, Bluetooth LoRaWAN Cellular 6G	LoRaWAN platform	Cloud Big Data Analytics	Blockchain technology	AI	-	-
[86]	-	-	-	-	ML methods	-	Domain radiative transfer models, future multi-Domain radiative transfer models (e.g., SCOPE) with dynamic crop growth models for agroecosystems DTs
[87]	Wireless communication technologies, Wi-Fi, Cellular	IoT dashboards	Alibaba cloud, Amazon web services, Microsoft Azure, Google Cloud platform, IBM Cloud—Cloud computing service providers	-	Blockchain	-	-
[88]	-	Predictive analytics.	-	Big Data	ML and AI algorithms to attain surgical control over all operational aspects of production activities. LED actuators	Simulation, analysis and prediction	Modelling/Simulating seed fertility, fertilizer, pesticides, pollution challenges/Soil agent (hydrological models, soil data), crop agent. Predictive control process models (heating, ventilation). Plant development modelling

## Data Availability

Any data presented in this study are available within the article.

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
