# Peer review of "Enhancing Smart Agriculture by Implementing Digital Twins: A Comprehensive Review"

_sensors, 2023, doi:10.3390/s23167128_

Round 1

Reviewer 1 Report

Review of

Enhancing Smart Agriculture by Implementing Digital Twins: A comprehensive review

by

N. Peladarinos, D. Piromalis, V. Cheimaras, E. Tserepas, R.A. Munteanu, and P. Papageorgas

1.       Line 50; it is stated that more informed decisions can be made about soil health. Does this incorporate the soil microbiome? A train of thought states that we would be better off maintaining the soil microbiome than using fertilizer. Would the author(s) being willing to address the soil microbiome more directly as a fairly new management concern?

2.       Line 69; it is assumed by the review that the smart systems are transformed to address the agriculture domain. The ‘and ‘should be removed.

3.       Lines 153-163; the reviewer sees data driven models as being based on the assumption that the past is a predictor of the future. The author(s) may want to say something about what data based models are good for and mention something about the effort required to develop data based models for a specific field like water flow through which would require many sensors and likely years of time to collect data covering a large range of inputs. The author(s) may also want to say something about how accurate are the models expected to be and mention uncertainty analysis to look at how errors compound from model to model and ways of reducing output sensitivity to errors in the models.

4.       Lines 167-170; isn’t the model based on probability? The system could be wrong even when the inputs are adequate…probability?

5.       Line 184; if the decision is optimized without farmer input, is the farmer no longer managing the farm?

6.       Line 217; how does security ensure optimum production performance? Please discuss.

7.       Lines 197-231; this implies that the CDT will know as an example that it got burned and thereby remember the circumstances that led to the burn. It may even observe and remember what caused the fire. Is that the same as being able to predict when the fire will start and how it will spread? The author(s) may want to address where the state of the art is and what its capabilities will be in five or ten years.

8.       Lines 247-253; if the CDT optimizes solutions, how does the farmer fit into the decision process? The reviewer assumes that changing practices would not require complete retraining of the AI, or does it?

9.       Line 299; change ‘have to be dried out, cooled down, transported’ to ‘have to be dried, cooled, transported’.

10.   Line 268; change ‘disease spread out’ to disease spread’.

11.   Lines 370-377; Would pollutants be monitored allowing one to see where nonpoint source water pollutions is from and how it spreads? What about air quality issues related to ag?

12.   Lines 443-450; it would be helpful if the author(s) expanded on what hazardous situations would be mitigated.

13.   Lines 452-460; the reviewer reads this section to discuss what could be not what is. The author(s) would help readers if they made it more clear what is current state of the art verse where it is or should go. Could the title be changed to reflect that it is a review paper and a vision paper of the future?

14.   Lines 559-582; reliability is discussed quite a bit. How reliable is the system and how accurate does it have to be? What sensor, actuator, data transmission system is without error and deviation. Every device is built within tolerances so two devices that are ‘identical’ at some level give different answers.

15.   Lines 686-707; validating models (data driven models more so) is always an issue. The reviewer sees that this may be more difficult if all the data is used to develop the data based model. What would the author(s) suggest using as the data for verification?

16.   Line 733; change ‘before the built’ to before the build’.

17.   Lines 744-749; the author(s) address the fact that the system will be inaccurate (likely not precision too). Thank you.

18.   Line 808; the term precision is used which to the reviewer means the result is repeatable to some degree, but it does not me that it is close to the accepted true value. Do the author(s) intend to address precision or accuracy?

19.   Line 812; if AI develops outcomes tailored to my likes, does that make it harder for me to change behavior and farming practices?

20.   Section 4; this is a difficult task and the reviewer would like to say that most comments are for discussion. Did the author(s) check to see if similar titles were because of follow on papers? Were the key words in the key word list, title, or paper?

21.   Lines 992-999; would programs like LabView fit in this discussion?

22.   Lines 1032-1112; thoughts that came up when reading are 1) it would be helpful if the author(s) added a table of acronyms, 2) A table listing the types of sensors that are currently used might help readers, and 3) what bandwidth is there with the communications systems and how long will it be before it is fully utilized?

23.   Line 1113-1114; the author(s) state that modelling relies on available data. This is true for data driven models, but not as true for analytical (physics) models. Agriculture (plant and animal sciences) has relied heavily on data driven models while other fields rely more on analytical driven models. Agriculture may be a good place for this type of thinking.

Reviewer 2 Report

My congratulations for this submitted article. It is one of the best review papers about this subject. It is well-researched and contains the important references. In my opinion it can be published as it is.

Reviewer 3 Report

The paper present a comprehensive overview of the contemporary state of research on digital twins in smart farming.

Overall, the paper is well-written, but I have only a few observations and questions:

-    from my point of view, I think it is better to include in the Materials and Methods chapter the part where you describe how you performed the search and how you selected the articles for review (section 4);

-     section 2, with Dt definition and applications in agriculture could be included in the Introduction or in a separate section;

-        it would be advisable to add in the introductory part a paragraph in which you review the review articles from the specialized literature that are closed with the topic and to highlight the novelty and own contribution of your article compared to the current scientific literature;

-   for the methodology part, did you use any established method for searching and selecting articles (like PRISMA)?

-      what eligibility criteria (inclusion and exclusion) did you use to choose the articles?

-      how many authors were involved in the article selection procedure?

-      the Conclusion section is missing.
